# Conformal Prediction for Semantically-Aware Autonomous Perception in Urban Environments

**Achref Doula**    **Tobias Güdelhöfer**    **Max Mühlhäuser**    **Alejandro Sanchez Guinea**

Telecooperation Lab
Technical University of Darmstadt, Germany
{doula, max, sanchez}@tk.tu-darmstadt.de

**Abstract:**

We introduce Knowledge-Refined Prediction Sets (KRPS), a novel approach that performs semantically-aware uncertainty quantification for multitask-based autonomous perception in urban environments. KRPS extends conformal prediction (CP) to ensure 2 properties not typically addressed by CP frameworks: semantic label consistency and true label coverage, across multiple perception tasks. We elucidate the capability of KRPS through high-level classification tasks crucial for semantically-aware autonomous perception in urban environments, including agent classification, agent location classification, and agent action classification. In a theoretical analysis, we introduce the concept of semantic label consistency among tasks and prove the semantic consistency and marginal coverage properties of the produced sets by KRPS. The results of our evaluation on the ROAD dataset and the Waymo/ROAD++ dataset show that KRPS outperforms state-of-the-art CP methods in reducing uncertainty by up to 80% and increasing the semantic consistency by up to 30%, while maintaining the coverage guarantees.

**Keywords:** Robot Perception, Uncertainty in Robotics, Semantics for Robotics

## 1   Introduction

In urban environments, autonomous systems are tasked with navigating complex and dynamic scenes, necessitating robust decision-making mechanisms that can interpret diverse elements with a high-level semantic understanding. To reason about the scene, autonomous systems perform a series of critical tasks, distinguishing between different types of agents like pedestrians and vehicles, understanding their actions such as walking, turning, or stopping, and identifying their locations relative to key infrastructure like crosswalks and intersections [1, 2, 3, 4, 5, 6]. Such tasks require the construction of semantic relationships between scene elements to ensure safety and enhance model interpretability [7, 8, 9]. However, the inherent unpredictability of urban environments and the safety-critical applications of autonomous perception in urban environments make it essential to integrate semantically-aware reasoning with uncertainty quantification. Combining semantic and uncertainty awareness allows autonomous agents to make reliable and contextually appropriate decisions, adapting to the urban landscape dynamically.

Conformal prediction (CP) [10, 11, 12] emerges as a versatile approach for uncertainty quantification in learning-based models. Compared to other uncertainty quantification frameworks in deep learning, such as Bayesian networks [13], Radial Basis Functions networks [14], and evidential learning [15], CP provides uncertainty-aware predictions by generating prediction sets that are theoretically guaranteed to include the correct class with user-defined confidence. The guarantees offered by CP-based frameworks ensure the reliability of the underlying systems, a property that is much needed to improve trust and large-scale adoption of autonomous systems interacting with urban environments. CP has been effectively applied in various areas of robotics, including robot manipu-

8th Conference on Robot Learning (CoRL 2024), Munich, Germany.

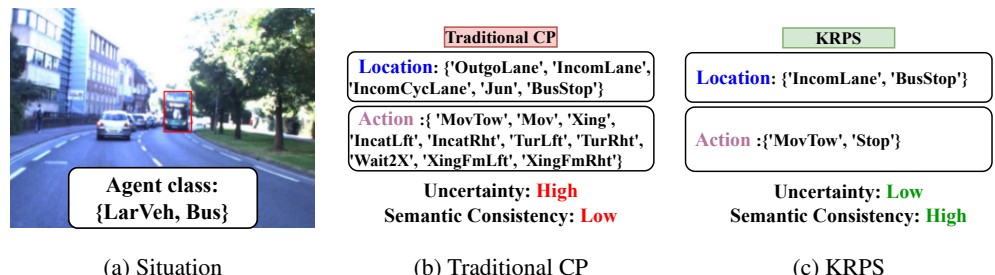

Figure 1: An example of refined sets using KRPS on a scene from the ROAD dataset [1]. KRPS reduces the location set size by 60% and the action set size by 80%. In addition to the set size reduction, the produced classes are semantically consistent with the predicted agent classes: large vehicle and Bus. More qualitative examples are presented in Appendix D.

lation [16], human-robot interaction [17, 18], autonomous perception [19, 20, 21], and autonomous navigation [22, 23].

The separate application of traditional CP to different tasks does not guarantee semantic consistency. In our context, semantic consistency refers to the logical alignment of predictions across different tasks, ensuring that the inferred attributes are contextually appropriate and mutually compatible. To illustrate this, we present in Figure 1b the result of one of our experiments on the ROAD dataset [1], where an autonomous agent uses three distinct models to classify an agent, its location, and its action. Applying traditional CP to each task in isolation results in high uncertainty and low semantic consistency. High uncertainty is shown by the numerous potential classes of actions and locations. Low semantic consistency is seen in the conflicting locations (e.g., *incoming cycle lane*) and actions (e.g., *crossing*) for the agent classes (i.e., *large vehicle* and *bus*). It is important to note that simply removing inconsistent classes compromises CP coverage guarantees, which is undesirable.

In this work, we address the construction of conformal prediction sets across various classification tasks for perception, such as agent, action, and location classification, while ensuring conformal prediction validity and semantic consistency among the produced sets for each task. We propose Knowledge-Refined Prediction Sets (KRPS), a novel CP approach that reduces uncertainty in autonomous agents in urban environments by producing valid, uncertainty-aware prediction sets that are semantically consistent across multiple tasks. KRPS consists of a knowledge graph that links the different tasks, and a sequential[1] set construction procedure based on an intial task to ensure the produced sets for subsequent tasks cover the true label with a user-defined probability. The sequential processing of tasks in KRPS allows for efficient on-demand refinement of subsequent task predictions, based on initial task results. As depicted in Figure 1c, KRPS sequentially refines the prediction sets for the location and action tasks using the prediction sets from the agent classification task and a knowledge graph. In this case, KRPS leads to more semantically consistent results, reflected by the alignment between the agent classes, its possible locations, and actions, and leads to 80% lower uncertainty when compared to traditional CP, reflected by the lower number of classes in the prediction sets. In cases where the knowledge graph is incomplete, KRPS manages corner cases, such as anomalies where vehicles are located on sidewalks, by outputting empty prediction sets, indicating the absence of semantically consistent labels. We provide a theoretical analysis of KRPS, introducing the concept of conditional semantic consistency for multitask-based autonomous perception and demonstrating the semantic consistency and marginal coverage properties of the produced sets. We evaluate KRPS on 2 large-scale autonomous driving datasets, namely the ROAD dataset [1] and Waymo/ROAD++ dataset [24], from the scope of 3 high-level perception tasks that are crucial for semantically-aware autonomous perception and high-level reasoning in urban settings: object class, location, and action. The results of our evaluation show that KRPS outperforms state-of-the-art conformal prediction approaches in reducing the uncertainty of the models by up to 80% and increasing the semantic consistency by up to 30%, through the generation of smaller prediction sets that are semantically consistent through the tasks while holding the coverage guarantees.

---

[1]Sequentiality in our context means refining predictions for one task at a time.

## 2 Background on Conformal Prediction

We focus on CP for classification, as it is the relevant task for our work. Let $g_\theta$ be a classifier pre-trained on a dataset $D_{train}$. The model $g_\theta$ outputs estimated probabilities for each class (e.g., softmax scores), such that $g_\theta(X) \in [0, 1]^K$, where $X$ is an input image and $K$ is the number of classes considered.

**Assumption.** CP assumes the existence of *exchangeable* unseen pairs of data examples that are drawn from the same data distribution as $D_{train}$, to which we refer to as calibration set $D_{cal}$. The goal of CP is to construct prediction sets for unseen data samples $C(X_{test})$ that are valid, i.e., they contain the true label $Y_{test}$ with a probability $1 - \alpha$, where $\alpha$ is a user-defined error rate. This property is called *marginal coverage* and is expressed in Equation 1.

$$\mathbb{P}[Y_{test} \in C(X_{test})] \geq 1 - \alpha \tag{1}$$

To construct the prediction sets, CP requires a test statistic called *non-conformity score* $S(X, Y)$, which measures the dissimilarity between predictions generated for an unseen data point and the training data. Based on $S$, the elements of $D_{cal}$ are ranked, and the empirical $1 - \alpha$ quantile $\hat{q}$ is calculated. For a novel $X_{test}$, for which $Y_{test}$ is unknown at test time, $C(X_{test})$ is constructed as $C(X_{test}) = \{Y : S(X, Y) \leq \hat{q}\}$. This procedure can be expressed through p-values that measure the probability of non-conformity, as formalized in Theorem 2.1.

**Theorem 2.1 (Conformal Prediction [10])** *Let $D_{train}$, $D_{cal}$, and $X_{test}$ be sets of exchangeable random variables. Let S be a non-conformity score and $\alpha$ be a user-defined error rate. The set $C(X_{test})$ defined as $C(X_{test}) = \{Y \in \mathcal{Y} : p\text{-value}(S(X_{test}, Y)) \geq \alpha\}$ satisfies the marginal coverage property stated in Equation 1.*

## 3 Knowledge-Refined Prediction Sets

We address the problem of autonomous perception for urban scene understanding based on the reasoning across multiple high-level tasks, that allow the agent to make uncertainty-aware and semantically consistent predictions. We focus on 3 classification tasks: agent type, agent location, and agent action, that are deemed necessary to achieve a high-level semantic understanding of the robot's environment [1, 2, 3, 4]. We assume the existence of 3 separate neural networks $g^c$, $g^l$, and $g^a$ that output the agent class $\hat{Y}_j^c \in \mathcal{Y}^c$, agent location class $\hat{Y}_j^l \in \mathcal{Y}^l$, and agent action class $\hat{Y}_j^a \in \mathcal{Y}^a$, respectively, given a bounding box around the agent of interest $X_j$, where $X_j \in \mathcal{X}$, and $\mathcal{X}$ represents the set of all agents of interest in a scene. Given the models $g^c$, $g^l$, and $g^a$, a calibration set $D_{cal}$, and an unseen test sample $X_{test}$, our goal is to sequentially construct the prediction sets $C_l(X_{test})$ for location classification, and $C_a(X_{test})$ for action classification, given a $C_c(X_{test})$ that was generated by an agent classification CP algorithm. We opt for sequential task refinement as it permits on-demand processing of subsequent tasks in case of need. We desire our generated sets to have two properties. The first property is marginal coverage as described in Equation 1, which means that the predicted sets are guaranteed to include the true label $Y_{test}$ with a probability of at least $1 - \alpha$. The second property is semantic consistency, which means that the prediction sets for each of the classification tasks need to account for classes included in other sets. For example, for $C_c(X_{test}) = \{\text{car}, \text{medium vehicle}, \text{large vehicle}\}$ it is clear that $Y_{test}$ corresponds to a vehicle, and so $C_l(X_{test})$ should not include locations that are not proper to vehicles, such as *sidewalk*. Traditional CP set-ups do not guarantee these properties, as they only consider one task. In this work, we propose knowledge-refined prediction sets (KRPS), a CP-based approach that constructs the prediction sets described above given an initial task and a knowledge graph. For the remainder of this paper, we refer to sets constructed by KRPS by $C_c^{KRPS}(X_{test})$, $C_l^{KRPS}(X_{test})$, and $C_a^{KRPS}(X_{test})$.

KRPS consists of two components: a knowledge graph and a knowledge-based multitask conformal prediction algorithm. The knowledge graph models the semantic relationships between the different tasks we consider. For a particular agent class, only a subset of all possible locations and actions can

be assigned. The other component of our approach is the knowledge-refined multitask conformal prediction algorithm that leverages the knowledge graph to sequentially construct prediction sets for each task, that are semantically consistent while satisfying the coverage guarantees. The semantic consistency is ensured through a knowledge-based pruning procedure that removes location and action classes if they are semantically inconsistent with the agent class.

**Knowledge Graph Construction.** To construct knowledge-refined prediction sets (KRPS), we build a knowledge graph $\mathcal{K}$ in order to model the semantic relationships between the tasks. The knowledge graph in our implementation draws from common sense and available labels in large-scale datasets for urban scene understanding, such as the ROAD and Waymo/ROAD++ datasets [1, 24], as we detail in Appendix B. Using the labels for each task, the semantic relationships between the tasks are constructed as follows. Given a task $\mathcal{T}_c$ and $\mathcal{T}_a$ referring to the tasks of agent classification and agent action classification, $\mathcal{K}$ contains a set-valued deterministic function $\mathcal{M}_{c \to a} : \mathcal{Y}^c \to 2^{\mathcal{Y}^a}$ that maps agent class labels to the set of all possible action labels, as it is depicted in Figure 2a. Figure 2b shows an example of this mapping, where the class cyclist is mapped to all its possible locations by $\mathcal{M}_{c \to l}$, and to all its possible actions by $\mathcal{M}_{c \to a}$. Given $\mathcal{K}$, we introduce the notion of conditional semantic consistency between prediction sets for each task.

**Definition 3.1 (Conditional Semantic Consistency)** *Let $C_j$ and $C_i$ be 2 prediction sets containing class candidates for the tasks $\mathcal{T}^j$ and $\mathcal{T}^i$, respectively. $C_i$ is semantically consistent **with respect to** $C_j$ and $\mathcal{K}$, if:*

$$\forall Y_i \in C_i, \exists Y_j \in C_j | Y_j \in \mathcal{M}_{i \to j}(Y_i) \tag{2}$$

Equation 2 implies that every element in $C_i$ has a pre-image in $C_j$ by the semantic mapping $\mathcal{M}_{i \to j}$. Note that with this definition, $C_j$ can have elements that do not have images in $C_i$.

**KRPS Construction Procedure.** KRPS is based on sequentially pruning prediction sets using knowledge from $\mathcal{K}$. For a given scene and an object $X_{test}$, we start by constructing the prediction set for the agent class task $C_c^{KRPS}(X_{test})$. Since our approach is sequential, the construction procedure of the prediction set for the first task follows the usual CP procedure based on Theorem 2.1. Next, we acquire all possible locations of agent classes in $C_c^{KRPS}(X_{test})$ using the semantic mapping from the knowledge graph $\mathcal{K}$. The set of all possible locations that correspond to $C_c^{KRPS}(X_{test})$ is $C_l^{\mathcal{K}} = \mathcal{M}_{c \to l}(C_c^{KRPS})^2$. We use $C_l^{\mathcal{K}}$ as a starting set for further pruning, instead of considering the full space of labels $\mathcal{Y}^l$. The pruning consists of a conformal prediction step based on a conformity score proper to the location classification task. Examples of such conformity scores are $1 - Softmax$. The conformal prediction step produces the set $C_l^{KRPS}$. Applying the steps described above results in a multi-hypotheses testing problem, where the family-wise error increases. To account for the MHT

---

**Algorithm 1:** KRPS Procedure

**Input** : $\mathcal{K}$: knowledge graph.
$\mathcal{T}_1..\mathcal{T}_m$: set of tasks.
$\mathcal{M}_{\mathcal{T}_i \to \mathcal{T}_{i+1}}$, semantic mappings.
$M$: correction procedure.
$D_{cal}$: calibration set.
$\alpha$: user-defined error rate.

**Output:** $C_{\mathcal{T}_i}^{KRPS}, i \in [1, m]$: semantically consistent prediction sets.

1  $C_{\mathcal{T}_0}^{KRPS} \leftarrow \cdot \mathcal{Y}^0$
2  $P_{\mathcal{T}_i} \leftarrow 1, \forall Y^{\mathcal{T}_i} \in \mathcal{Y}^{\mathcal{T}_i}, i \in [0, m]$
3  **for** $i = 1$ *to* $m$ **do**
4      $C_{\mathcal{T}_i}^{\mathcal{K}} \leftarrow \cdot \mathcal{M}_{\mathcal{T}_{i-1} \to \mathcal{T}_i}(C_{\mathcal{T}_{i-1}}^{KRPS})$
5      $C_{\mathcal{T}_i}^{KRPS} \leftarrow \{\}$
6      **for** $y \in C_{\mathcal{T}_i}^{\mathcal{K}}$ **do**
7          $P_{\mathcal{T}_i} \leftarrow M(\text{p-value}(X_{test}, D_{cal}))$
8          **if** $P_{\mathcal{T}_i} \geq \alpha$ **then**
9              $C_{\mathcal{T}_i}^{KRPS} \leftarrow C_{\mathcal{T}_i}^{KRPS} \cup \{y\}$
10     **end for**
11 **end for**
12 **return** $C_{\mathcal{T}_i}^{KRPS}, i \in [1, m]$

---

problem, applying a correction procedure $M$ of the p-values is required, as we detail in Appendix A. Prediction sets for further tasks can be acquired by applying the same procedure sequentially using $C_l^{KRPS}$. Using the procedure explained above, the generated sets are guaranteed to be semantically consistent and to have coverage as formalized in Equation 1. The KRPS set construction proce-

---

[2]$\mathcal{M}_{c \to l}$ is a point to set mapping. We abuse the notation to indicate the set of images for every elements in $C_c^{KRPS}$ using the mapping $\mathcal{M}_{c \to l}$.

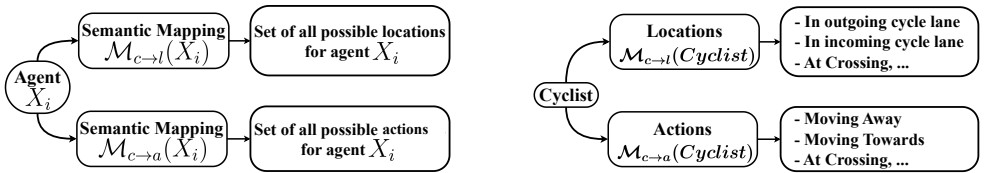

(a) General structure of the knowledge graph.   (b) Knowledge graph for the agent class *Cyclist*.

Figure 2: Knowledge-graph structure used in KRPS. Figure 2a shows the structure of the knowledge graph with the semantic mappings $\{agent \rightarrow location\}$ and $\{agent \rightarrow action\}$. Figure 2b depicts an example graph for the agent class *Cyclist*. The full construction procedure is in Appendix B.

dure is formalized in Theorem 3.1 and described in Algorithm 1. The proof of Theorem 3.1 and its corollary, along with an analysis of their practical implications, are provided in Appendix A.

**Theorem 3.1 (Knowledge-Refined Conformal Prediction)** *Given 2 tasks $\mathcal{T}_c$ and $\mathcal{T}_l$, the semantic mapping function $\mathcal{M}_{c \rightarrow l}$, and the prediction sets $C^c(X_{test})$ and $C^l(X_{test})$, the set $C_l^{KRPS}(X_{test}) = C^l(X_{test}) \cap \mathcal{M}_{c \rightarrow l}(Y_{test}^c)$ is **semantically consistent** with respect to $C_c^{KRPS}$ and $\mathcal{K}$, and satisfies the **marginal coverage** described in Equation 3.*

$$\mathbb{P}[Y_{test} \in C_l^{KRPS}(X_{test})] \geq 1 - \alpha \tag{3}$$

We prove this property by leveraging the probability conservation of the deterministic semantic mappings between the task and proposing a generalizing of the cascaded CP Theorem [25] to cover multitask settings. We present the detailed proof of Theorem 3.1 in Appendix A.

**Corollary 3.1 (Transitivity of conditional semantic consistency)** *Let $C_c^{KRPS}$, $C_l^{KRPS}$ and $C_a^{KRPS}$ be 3 prediction sets constructed using KRPS. If $C_l^{KRPS}$ is semantically consistent, with respect to $C_c^{KRPS}$ and $\mathcal{K}$, and $C_a^{KRPS}$ is semantically consistent, with respect to $C_l^{KRPS}$ and $\mathcal{K}$, then and $C_a^{KRPS}$ is semantically consistent, with respect to $C_c^{KRPS}$ and $\mathcal{K}$.*

We prove the transitivity property by leveraging the semantic consistency between prediction sets of subsequent tasks, guaranteed by Theorem 3.1. We present a proof of Corollary 3.1 in Appendix A.

## 4 Evaluation

To support the theoretical results for guaranteed coverage and semantic consistency across different tasks executed sequentially, we evaluate KRPS on 2 large-scale urban scene understanding datasets. We compare the performance of KRPS against 3 baselines including 2 state-of-the-art approaches, using 2 widely used CP metrics: deviation from coverage rate and set size. Since we are the first to consider semantic consistency across tasks in CP, we introduce a metric for semantic consistency.

### 4.1 Datasets

**The ROAD Dataset [1].** The Road Event Awareness dataset (ROAD) for autonomous driving includes 22 videos, and 1.7M labels across 11 agent classes, 15 locations, and 23 actions under various weather conditions, providing a diverse and challenging set of scenarios for testing the effectiveness of KRPS. ROAD offers (agent, location, action) triplets, ideal for a comprehensive evaluation.

**The Waymo/ROAD++ Dataset[3].** The Waymo/ROAD++ dataset, an extension of the ROAD dataset, incorporates 3.5M agents, 4.3M actions, and 4.2M locations from the expansive Waymo autonomous driving dataset [24], annotated with road events akin to those in the ROAD dataset.

**Implementation.** For the evaluation, we use the 3D-RetinaNet [1] with separate heads for each classification task, outputting softmax scores. The model is trained on each dataset for 30 epochs

---

[3]Waymo/ROAD++: `https://sites.google.com/view/road-plus-plus/dataset`

with an Adam optimizer at a learning rate of $lr = 0.001$. We randomly divide the validation splits of both datasets in 3 equal-sized portions as follows: *eval split 1* for validation, *eval split 2* for calibration, and *eval split 3* for testing. All experiments use Bonferroni correction for prediction robustness, with results averaged over 100 trials on a single 24GB RTX 3090.[4]

## 4.2 Metrics

**Deviation from Target Coverage (DTC).** The deviation from the target coverage rate (DTC) quantifies the discrepancy between the achieved coverage, $Cov_{approach}$, and the desired target coverage, $Cov_{target} = 1 - \alpha$, as defined in Equation 4.

$$DTC = Cov_{approach} - Cov_{target} \tag{4}$$

Negative DTC values indicate coverage below $1 - \alpha$, indicating a failure to meet the theoretical lower bound. DTC values nearing zero suggest that the actual coverage closely aligns with the target coverage. **Smaller positive** values denote accurate coverage rates above the desired threshold.

**Set Size (SS).** We measure the average set size provided by KRPS. As providing the full set of labels would be a trivial output for CP-based methods to achieve the coverage guarantees, a smaller set size demonstrates better predictive efficiency.

**Conditional Semantic Consistency (SC).** We introduce a novel metric to measure the semantic consistency between the generated sets predicted for the task $\mathcal{T}^i$ with respect to $\mathcal{T}^{i-1}$ and $\mathcal{K}$. For a prediction set $C_{\mathcal{T}^i}$, semantic consistency represents the rate of semantically consistent classes in the entire set and is formulated in Equation 5. A higher semantic consistency value demonstrates better coherence between the prediction sets across the tasks.

$$SC(C_{\mathcal{T}^i}|C_{\mathcal{T}^{i-1}}, \mathcal{K}) = \frac{|\{Y : Y \in C_{\mathcal{T}^i} \cap \mathcal{M}_{\mathcal{T}_{i-1} \to \mathcal{T}_i}(C_{\mathcal{T}_{i-1}})\}|}{|C_{\mathcal{T}^i}|} \tag{5}$$

## 4.3 Baselines

Inspired by recent CP works, such as [26, 25], we evaluate KRPS with the Least Ambiguous Set Classifier (LAC), for which the scoring function is $1 - Softmax$ [27], and 2 state-of-the-art CP scores. The first approach is adaptive predictive sets (APS) [28], which is a scoring technique known to improve conditional coverage. The second approach is regularized adaptive predictive sets (RAPS) [29], which is known to generate notably smaller predictive sets. RAPS stands unparalleled in minimizing the size of prediction sets. Thus, it is crucial to explore the effectiveness of KRPS in further compressing the predictive set sizes relative to those generated by RAPS.

## 4.4 Tasks

Our evaluation considers 3 tasks on both datasets, namely agent, location, and action classification. Since our approach acts sequentially on multiple tasks, the first task is not refined, which makes the construction of the prediction set for the first task (i.e., agent classification), coincide with the underlying CP process. The task sequences we consider are: $\{agent \to location\}$ and $\{agent \to action\}$. To conduct our evaluation, we first construct the prediction sets for the agent classification using $Softmax$, APS, and RAPS. Using the predicted agent class sets, we report the results for the subsequent tasks: location classification, and action classification. In Appendix C, We report the results of longer task sequences using 3 tasks to show the capability to handle longer sequences of tasks with different orders.

## 4.5 Comparison with State-of-the-Art

We conduct our evaluation using different values of $\alpha$ ranging from 0.1 to 0.5 with a step of 0.1. We present the results on the ROAD and Waymo/ROAD++ datasets in Table 1a and Table 1b, re-

---

[4]Our code is available here: https://gitlab.com/achref.d/krps.

| Task | Score | Method | α = 0.1 | | | α = 0.2 | | | α = 0.4 | | |
|---|---|---|---|---|---|---|---|---|---|---|---|
| | | | SS | DTC | SC | SS | DTC | SC | SS | DTC | SC |
| Location | Softmax | Standard | 7.07 | 0.05 | 0.81 | 6.04 | 0.10 | 0.79 | 4.51 | 0.19 | 0.77 |
| | | KRPS | **5.86** | **0.02** | **1.00** | **4.93** | **0.05** | **1.00** | **3.75** | **0.09** | **1.00** |
| | APS | Standard | 7.12 | 0.05 | 0.80 | 6.07 | 0.10 | 0.79 | 4.61 | 0.20 | 0.76 |
| | | KRPS | **5.90** | **0.02** | **1.00** | **4.96** | **0.05** | **1.00** | **3.65** | **0.12** | **1.00** |
| | RAPS | Standard | 2.66 | 0.07 | 0.89 | 2.03 | 0.15 | 0.93 | 1.56 | 0.34 | 0.96 |
| | | KRPS | **2.18** | **0.05** | **1.00** | **1.78** | **0.13** | **1.00** | **1.45** | **0.13** | **1.00** |
| Action | Softmax | Standard | 7.48 | 0.01 | 0.79 | 5.98 | 0.01 | 0.78 | 3.77 | 0.06 | 0.77 |
| | | KRPS | **5.83** | **0.00** | **1.00** | **4.70** | **0.00** | **1.00** | **3.06** | **0.00** | **1.00** |
| | APS | Standard | 9.77 | 0.05 | 0.75 | 8.01 | 0.10 | 0.73 | 5.72 | 0.30 | 0.70 |
| | | KRPS | **7.29** | **0.02** | **1.00** | **6.01** | **0.03** | **1.00** | **4.35** | **0.11** | **1.00** |
| | RAPS | Standard | 4.67 | 0.07 | 0.84 | 3.94 | 0.13 | 0.87 | 2.54 | 0.25 | 0.92 |
| | | KRPS | **3.82** | **0.03** | **1.00** | **3.26** | **0.02** | **1.00** | **2.22** | **0.15** | **1.00** |

(a) Results on the ROAD dataset for for $\alpha = [0.1, 0.2, 0.4]$. **Bold** designates better performance.

| Task | Score | Method | α = 0.1 | | | α = 0.2 | | | α = 0.4 | | |
|---|---|---|---|---|---|---|---|---|---|---|---|
| | | | SS | DTC | SC | SS | DTC | SC | SS | DTC | SC |
| Location | Softmax | Standard | 8.10 | 0.05 | 0.80 | 6.04 | 0.11 | 0.73 | 4.13 | 0.21 | 0.72 |
| | | KRPS | **6.02** | **0.01** | **1.00** | **5.42** | **0.03** | **1.00** | **3.07** | **0.08** | **1.00** |
| | APS | Standard | 9.02 | 0.07 | 0.79 | 7.92 | 0.10 | 0.76 | 4.03 | 0.22 | 0.71 |
| | | KRPS | **7.50** | **0.01** | **1.00** | **5.99** | **0.06** | **1.00** | **3.24** | **0.14** | **1.00** |
| | RAPS | Standard | 2.45 | 0.09 | 0.87 | 2.07 | 0.15 | 0.91 | 1.83 | 0.34 | 0.96 |
| | | KRPS | **2.01** | **0.04** | **1.00** | **2.04** | **0.10** | **1.00** | **1.17** | **0.09** | **1.00** |
| Action | Softmax | Standard | 9.32 | 0.06 | 0.80 | 7.20 | 0.08 | 0.78 | 5.00 | 0.15 | 0.81 |
| | | KRPS | **6.09** | **0.01** | **1.00** | **4.17** | **0.03** | **1.00** | **3.50** | **0.10** | **1.00** |
| | APS | Standard | 10.41 | 0.05 | 0.77 | 6.84 | 0.13 | 0.74 | 5.36 | 0.12 | 0.78 |
| | | KRPS | **5.98** | **0.02** | **1.00** | **4.18** | **0.02** | **1.00** | **3.87** | **0.12** | **1.00** |
| | RAPS | Standard | 4.36 | 0.06 | 0.87 | 3.03 | 0.15 | 0.89 | 1.50 | 0.24 | 0.94 |
| | | KRPS | **3.57** | **0.04** | **1.00** | **2.88** | **0.02** | **1.00** | **1.07** | **0.07** | **1.00** |

(b) Results on the Waymo/ROAD++ dataset for for $\alpha = [0.1, 0.2, 0.4]$. **Bold** designates better performance.

Table 1: Results of our experiments. Based on the prediction sets for the agent classification task, prediction sets for the location and action are sequentially inferred using KRPS with 3 scoring functions: $Softmax$, APS, and RAPS. The standard mode refers to the baseline without KRPS.

spectively. Due to space constraints, we only report the results for $\alpha = [0.1, 0.2, 0.4]$, in the main manuscript. Results for the full set of $\alpha$ values and different task sequences are in Appendix C.

The results show that KRPS substantially reduces the set size for all underlying scores. For the location classification task, the average set size reduction is by $17\%$ for softmax, $18\%$ for APS, and $10\%$ for RAPS, for the ROAD dataset, and by $20\%$ for softmax, $20\%$ for APS, and $27\%$ for RAPS, for the ROAD++ dataset. For the action classification task, the set size reduction achieved by KRPS is more apparent and equals $24\%$ for softmax, $25\%$ for APS, and $16\%$ for RAPS, for the ROAD dataset. For the Waymo/ROAD++ dataset, the results show that KRPS achieved a set size reduction by $36\%$ for softmax, $38\%$ for APS, and $15.4\%$ for RAPS. The contribution of KRPS to set size reduction is particularly apparent for APS, in both datasets. The reason is that APS is built to achieve class-wise coverage to account for unbalanced classes in the dataset, which has the effect of increasing the set size. The application of KRPS succeeds in reducing the set size while holding the marginal coverage guarantees. KRPS holds the theoretical coverage guarantees for all values of $\alpha$ and for all the non-conformity scores, which is indicated by positive DTC values. Furthermore, KRPS adjusts the coverage rate for all scoring functions, leading to lower DTC values.

KRPS holds conditional semantic consistency guarantees as well, achieving $100\%$ of conditional semantic consistency in all experimental conditions. Compared to the baselines, KRPS increases the conditional semantic consistency in the task of location classification by an average of $26\%$ for softmax, $23\%$ for APS, and $6\%$ for RAPS, for the ROAD dataset, and by $26\%$ for softmax, $23\%$ for APS, and $6\%$ for RAPS, for the ROAD++ dataset. For the task of action classification on the ROAD dataset, KRPS increased the semantic consistency by $25\%$ for softmax, $24\%$ for APS, and $8\%$ for RAPS. For the Waymo/ROAD++ dataset as well, KRPS increases the semantic consistency rates for action prediction by $25.5\%$ for softmax, $31\%$ for APS, and $11\%$ for RAPS. We present more experiments on longer task sequences, the influence of the knowledge graph size, and the percentage of the empty sets in Appendices B, and E, respectively.

# 5   Discussion

Recent studies have underscored the importance of incorporating semantics [30, 31, 32, 33] and uncertainty [34, 35, 36, 37] into robotic decision-making frameworks, particularly as Large Language Models (LLMs) are increasingly used to enhance autonomous navigation [38, 39, 40, 41]. These models, prone to uncertainty and hallucinations [42], necessitate a detailed contextual understanding of the scene for effective operation [43]. Incorporating semantic awareness and uncertainty awareness enhances both the reliability and interpretability of these systems, addressing a fundamental challenge in their widespread adoption: trust [44, 45]. KRPS introduces a novel approach that combines these aspects to improve the overall efficacy of autonomous decision-making processes. KRPS represents the first approach that extends CP to a multitask learning setup without specific requirements regarding the nature or sequencing of tasks. The theoretical analysis and the empirical evaluation show that KRPS significantly reduces uncertainty, while ensuring semantic consistency, in dynamic urban environments. Furthermore, KRPS benefits from the inherent practicalities of CP, eliminating the need for continual model retraining. Instead, KRPS requires only a computationally efficient calibration process to integrate new data, assuming data exchangeability as per CP norms.

**Limitations.** As we see KRPS as a practical approach to consider uncertainty and semantic awareness in autonomous robots and systems, it is important to consider the limitations of its application. KRPS, being a CP-based approach, traditionally relies on the assumption of data exchangeability. This assumption can limit its effectiveness with scoring functions that fail to address out-of-distribution (OOD) data scenarios. To mitigate this, KRPS has been designed to be agnostic to the scoring function, as demonstrated in our empirical analysis, making it compatible with OOD-aware calibration procedures, such as [46, 47, 12]. The effectiveness of KRPS is also contingent on the integrity of the underlying knowledge graph. Ambiguities within the knowledge graph regarding class definitions can compromise prediction accuracy. To address this, our evaluation integrates ontological structures derived from the datasets utilized, such as ROAD and Waymo [1, 24], and aligns with established logical frameworks [2], facilitating more precise semantic interpretations. Moreover, KRPS is designed to identify and respond to these ambiguities by generating empty prediction sets to effectively manage scenarios not fully covered by the knowledge graph. This feature highlights cases requiring further scrutiny and ensures that KRPS maintains reliability despite incomplete ontologies. KRPS is a sequential approach, leveraging prediction sets from the initial task to generate semantically consistent sets for subsequent tasks. Its dependence on the quality of the first set is intrinsic. Although joint prediction set construction is viable [48], the sequential process allows for on-demand refinements exclusively for the relevant agents in the scene, leading to better efficiency.

**Practicality.** While our paper focuses on 3 high-level perception tasks for urban scene understanding, KRPS can be extended to other robot perception applications. The formulation of Algorithm 1 and Theorem 3.1 is adaptable to diverse environmental contexts. For example, KRPS can be applied to urban search and rescue, handling sequences of tasks under high uncertainty like victim detection, hazard identification, and robot action selection. Other use-cases can also be envisioned, such as robot manipulation, where a robot is tasked to recognize and sort objects of different sizes and put them in their corresponding places. KRPS provides uncertainty quantification and semantic consistency at each stage, ensuring accurate object classification, optimal grasp strategy selection, and precise object placement. We elaborate more on these applications in Appendix F.

# 6   Conclusion

We introduced KRPS, a Knowledge-Refined Prediction Set approach, tailored for multitask learning in urban scene understanding. By refining prediction sets through semantic relationships among tasks, KRPS ensures semantic consistency and adheres to coverage guarantees by producing significantly smaller prediction sets. KRPS shows promise for real-world applications, particularly in autonomous driving. Due to safety concerns, initial tests were limited to realistic, and challenging datasets. Future work should focus on integrating KRPS into operational systems, with rigorous safety validations and controlled real-world trials to ensure trustworthy autonomous navigation.

**Acknowledgments**

This work has been partially funded by the German Federal Ministry of Education and Research (BMBF) through the Software Campus Program.

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

# Supplementary Material for the Paper

## Conformal Prediction for Semantically-Aware Autonomous Perception in Urban Environments

## A  Proofs

### A.1  Background For Proofs: Cascaded Conformal Prediction

Cascaded conformal prediction (cascaded-CP) [25] is a technique that allows to prune prediction sets sequentially for a *single task*, using a cascade of different non-conformity scores over $m$ steps. Since different statistical tests are applied to the data set, the multiple hypothesis testing (MHT) problem arises, which leads to an increased family-wise error rate (i.e., false positives), making the CP procedure invalid. Cascaded-CP makes use of p-value correction procedures $M$, such as Simes corrections, to account for MHT problem. Cascaded-CP is formalized in Theorem A.1.

**Theorem A.1 (Cascaded-CP [25])** *For any sequence of non-conformity measures* $(S_1, ..., S_m)$, *which yields p-values* $(P_1, ..., P_m)$ *and* $\alpha \in [0, 1]$, *the prediction set* $C^j(X_{test})$ *at step* $j < m$ *is defined as:*

$$C^j(X_{test}) = \{Y \in \mathcal{Y} : \tilde{P}_j^y > \alpha\} \tag{6}$$

*where* $\tilde{P}_j^y$ *is the corrected p-values using the procedure* $M$ *for candidate* $y$ *at step* $j$. *Then* $\forall j \in [1, m]$, $C^j(X_{test})$ *satisfies Equation 1, and* $C^m(X_{test}) \subseteq C^j(X_{test})$.

In this work, we show that Theorem A.1 can be extended to cover multiple tasks in cascade. Furthermore, in our case the tasks are not required to share the same label space. By constructing a knowledge graph facilitating semantic mapping between tasks, our approach accommodates different tasks. Notably, Cascaded-CP can be seen as a sub-case of our work if all tasks are identical.

## A.2  Proof of Theorem 3.1

We consider 2 tasks $\mathcal{T}_c$ and $\mathcal{T}_l$ that are performed sequentially. Our goal is to prove that the prediction set $C_l^{KRPS}$ obtained by performing any CP procedure on the set $C_l^{\mathcal{K}}$, which represents the semantic mapping $\mathcal{M}_{c \to l}$ of the set $C_c^{KRPS}$, satisfies 2 properties: marginal coverage, and semantic consistency with respect to $C_c^{KRPS}$ and $\mathcal{K}$.

**Marginal Coverage**    First, we prove the marginal coverage property of the set $C_l^{KRPS}$, that is:

$$\mathbb{P}[Y_{test}^l \in C_l^{KRPS}(X_{test})] \geq 1 - \alpha \tag{7}$$

The set $C_c^{KRPS}$ is constructed using a CP procedure, meaning that it satisfies Equation 1, and we have:

$$\mathbb{P}[Y_{test}^c \in C_c^{KRPS}(X_{test})] \geq 1 - \alpha \tag{8}$$

The semantic mapping $\mathcal{M}_{c \to l}$ is a deterministic mapping that assigns a set of possible locations to each element of $C_c^{KRPS}(X_{test})$. Knowing that the true value of the subsequent task $Y_{test}^l$ is the image of the true value of the starting task $Y_{test}^c$, the resulting mapping set $C_l^{\mathcal{K}}(X_{test}) = \mathcal{M}_{c \to l}(C_c^{KRPS}(X_{test}))$ contains the $Y_{test}^l$ with a probability that is at least equal to $1 - \alpha$, as expressed in Equation 9.

$$\mathbb{P}[Y_{test}^l \in C_l^{\mathcal{K}}(X_{test})] \geq 1 - \alpha \tag{9}$$

From the CP coverage Theorem 2.1, we can conclude the existence of a p-value $P_l^{\mathcal{K}}$ that satisfies:

$$\mathbb{P}[Y_{test}^l \in C_l^{\mathcal{K}}(X_{test})] \geq 1 - \alpha \iff \mathbb{P}[P_l^{\mathcal{K}} \leq \alpha] \leq \alpha \tag{10}$$

At this step, we have two distinct p-values for the subsequent task $\mathcal{T}_l$, namely $P_l^{\mathcal{K}}$ and $P_l$. The p-value $P_l^{\mathcal{K}}$ is employed in the construction of the set $C_l^{\mathcal{K}}$, whereas $P_l$ is utilized for forming the set $C^l(X_{test})$, representing the CP-based set for task $\mathcal{T}_l$ independently of the knowledge provided by the prior task $\mathcal{T}_c$, established on $C_c^{KRPS}$.

Subsequently, we execute a cascaded-CP procedure using Theorem A.1 on the p-values $P_l, P_l^{\mathcal{K}}$. We incorporate a p-value correction procedure denoted as $M$, resulting in corrected p-values $\tilde{P}_l, \tilde{P}_l^{\mathcal{K}}$, and we have:

$$\mathbb{P}[Y_{test}^l \in C^l(X_{test}) \cap C_l^{\mathcal{K}}(X_{test})] \geq 1 - \alpha$$
$$\iff \mathbb{P}[Y_{test}^l \in C_l^{KRPS}(X_{test})] \geq 1 - \alpha$$

which is the result stated in Equation 3.

**Semantic Consistency**    Our goal now is to show that the newly constructed prediction set $C_l^{KRPS}$ is semantically consistent with respect to $C_c^{KRPS}$ and $\mathcal{K}$. This result of semantic consistency comes from the fact that for each element $Y^l$ of $C_l^{KRPS}$, we have $Y^l \in C^l(X_{test}) \cap \mathcal{M}_{c \to l}(C^c(X_{test}))$, meaning that $C_l^{KRPS} \subseteq \mathcal{M}_{c \to l}(C^c(X_{test}))$, which gives the semantic consistency property.

**Implications of Theorem 3.1**    The direct implication of Theorem 3.1 is that we can further refine the prediction sets generated by any CP procedure given $\mathcal{K}$ and a related task, by removing classes that are not semantically consistent with other tasks, without losing the property of marginal coverage, provided that the p-values are properly corrected. The semantic refinement in KRPS plays an additional role in highlighting corner cases. In urban applications, it is typical to start with a basic knowledge graph and incrementally add new class relationships as data becomes available. This process, however, may encounter corner cases or semantic inconsistencies, like vehicles on sidewalks, not covered by the knowledge graph. KRPS addresses these instances by outputting an empty prediction set when a semantically consistent vehicle position cannot be found. This empty

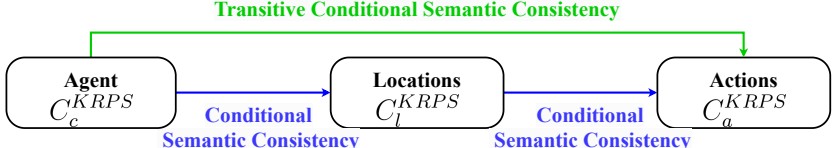

Figure 3: Implications of Theorem 3.1 (blue) and Corollary 3.1 (green) on the semantic consistency.

set signals potential knowledge graph gaps or corner cases, prompting further investigation by end-users. Updating the knowledge graph and performing a calibration step is sufficient to adapt to new data without retraining the model.

### A.3  Proof of Corollary 3.1

The sets $C_c^{KRPS}$, $C_l^{KRPS}$ and, $C_a^{KRPS}$ are 3 prediction sets constructed using KRPS, in the described order. Given this, our goal is to prove that $C_a^{KRPS}$ is semantically consistent with respect to $C_c^{KRPS}$ and $\mathcal{K}$. Since $C_a^{KRPS}$ is constructed based on Theorem 3.1, we have $C_a^{KRPS}$ is semantically consistent with respect to $C_l^{KRPS}$ and $\mathcal{K}$, i.e.,

$$\forall Y_a \in C_a^{KRPS}, \exists Y_l \in C_l^{KRPS}/Y_a \in \mathcal{M}_{l \to a}(Y_l) \tag{11}$$

Since $C_l^{KRPS}$ is constructed using KRPS based on Theorem 3.1, $C_l^{KRPS}$ semantically consistent with respect to $C_c^{KRPS}$ and $\mathcal{K}$ and we have:

$$\forall Y_l \in C_l^{KRPS}, \exists Y_c \in C_c^{KRPS}|Y_l \in \mathcal{M}_{c \to l}(Y_c) \tag{12}$$

Based on Equations 11 and 12, we can establish that:

$$\forall Y_a \in C_a^{KRPS}, \exists Y_c \in C_c^{KRPS}|Y_a \in \mathcal{M}_{c \to a}(Y_c) \tag{13}$$

Which gives that $C_a^{KRPS}$ is semantically consistent with respect to $C_c^{KRPS}$ and $\mathcal{K}$.

**Implications of Corollary 3.1**  Corollary 3.1 establishes the transitive properties of semantic consistency, as defined using Definition 3.1. As depicted in Figure 3, this result implies that two prediction sets from sequential tasks maintain semantic consistency, even if constructed independently, as long as they are built in sequence relative to a shared task in the middle. This finding holds significance as knowledge bases evolve over time with the inclusion of new tasks. Employing KRPS ensures guarantees of semantic consistency with all prior tasks, by uniquely verifying semantic consistency for the latest executed task.

**Comments on the Coverage Guarantees of KRPS.**  The coverage guarantee, as currently framed in Theorem 3.1, holds in expectation over the calibration sets, reflecting the standard practice in conformal prediction to ensure broad applicability and robustness across various operational scenarios [29]. KRPS prioritizes generalizability and statistical validity across diverse conditions, which is why the guarantee is designed to hold in expectation rather than conditionally. However, KRPS can be easily adapted to conditional coverage guarantees by adopting class conditional calibration procedures, such as in [49].

**Comments on the Semantic Consistency Guarantees of KRPS.**  KRPS guarantees conditional semantic consistency on the prediction sets of the subsequent tasks relative to the first task and the knowledge graph. This means that while KRPS ensures that the sets are semantically consistent given the output of the first task and according to the relationships defined in the knowledge graph, this consistency is conditional on those of initial outputs of the first task executed.

### A.4 Details About the Correction Procedure

In our implementation of KRPS, addressing the challenges of multiple hypothesis testing is crucial due to the simultaneous consideration of multiple semantic categories in the knowledge graph. To ensure the validity of our statistical inferences while managing the family-wise error rate (FWER), we employ the Bonferroni correction procedure. The Bonferroni procedure [50] is a conservative approach that adjusts for multiple comparisons by dividing the desired significance level $\alpha$ by the number of hypotheses tested, $m$.

Specifically, if individual tests are conducted at a significance level $\alpha$, the Bonferroni correction modifies the significance threshold for each test to $\frac{\alpha}{m}$. For KRPS, where each agent's classification can lead to multiple related hypotheses about actions and locations, the correction ensures that the overall probability of making one or more type I errors does not exceed $\alpha$.

Mathematically, this is represented as:

$$\alpha_{\text{adjusted}} = \frac{\alpha}{m}$$

Where $m$ is the total number of hypotheses, which corresponds to the number of potential action and location classifications linked to each agent class. By applying the adjustment for the values of $\alpha$, KRPS maintains the control over the error rates across all tests.

## B   Structure of the Knowledge Graph

In the following, we provide more details about the knowledge graph used to model the semantic relationships between the entities in the urban environment. We adopt a simple, yet effective ontological model based on the following semantic relationship: *Agent* performs *action* in *location*. It is possible to adopt different task orders, e.g., *Action* is performed in *location* by *agent*. In our setup, the classification of an agent, its location, and its action, correspond to different tasks, that are performed by separate models or separate heads of a single model. This is to ensure that we consider a multitask setup, in contrast to situations where a model outputs a triplet, which we consider as a case of multi-class classification tasks.

We adopt the class labels provided by the ROAD and the Waymo/ROAD++ datasets[5] [1, 24] for all the tasks that we consider. For Completeness, we report the list of agent, location, and action classes, as they are described in the ROAD dataset, in Table 2, Table 3, and Table 4, respectively.

Based on the class labels for each task, the semantic mapping functions $\mathcal{M}_{\mathcal{T}_i \to \mathcal{T}_j}$, where $T_i$ and $T_j$ represent the start and the subsequent tasks, respectively, are constructed through the examination of possible label assignments between the tasks in the training set.

More in details, the construction of the knowledge graph begins by loading semantically coherent triplets from ground truth labels in the training set, which define relationships between agents, actions, and locations. Each agent class is mapped to corresponding action and location classes through boolean arrays within a structured dictionary. The construction procedure ensures also that the reverse mappings can be queried, in case a different sequence task is considered. The knowledge graph construction procedure is detailed in Algorithm 2.

In the KRPS framework, the ground truth data is utilized not merely as examples for training models but also as a fundamental component for constructing the knowledge graph. This dual utilization allows the ground truth data to both train the predictive model and guide the development of semantically aware prediction sets using KRPS, afterwards, ensuring that the prediction sets are statistically valid and semantically consistent across the tasks. Under the assumption of data exchangeability, a central tenet of conformal prediction, we assume that the data in the training, calibration, and test sets come from the same distribution. This assumption helps limit the incompleteness of the knowledge graph construction procedure. Even if the knowledge from the training set is incomplete, KRPS

---

[5]Waymo/ROAD++: `https://sites.google.com/view/road-plus-plus/dataset`

manages this by outputting empty sets, prompting the user to address gaps in the knowledge graph or to identify potential anomalies from out-of-distribution samples.

It is important to acknowledge that the construction of knowledge graphs can be approached through various methodologies beyond the one implemented in KRPS. The method described herein was selected for its simplicity and effectiveness within our application scope. Alternative methods, such as [51, 52], demonstrate the capacity to construct complex knowledge graphs tailored to autonomous driving scenarios using different types of input data and for various tasks, such as perception and navigation. Such versatility not only illustrates the scalability of knowledge graph construction processes but also supports their adaptability across different semantic dimensions.

---

**Algorithm 2:** Construction of the Knowledge Graph

---

**Input** : $GT_{file}$:Ground truth annotations containing agent, action, and location classes from training set.

**Output:** $\mathcal{K}$: A dictionary representing the knowledge graph where agent classes are linked to semantically coherent actions and locations.

1: **procedure** BUILDKNOWLEDGEGRAPH
2:     Load ground truth labels from $GT_{file}$
3:     Extract semantically coherent triplets of agents, actions, and locations
4:     Initialize an empty dictionary for the knowledge graph
5:     For each agent class:
6:         Create a sub-dictionary for actions and locations
7:         Initialize boolean arrays for actions and locations based on class counts
8:     Populate the boolean arrays using triplet data:
9:         For each triplet:
10:             Mark the corresponding action and location as semantically coherent
11:     Return the constructed knowledge graph, $\mathcal{K}$
12: **end procedure**

---

### B.1  Complete Structure of the Knowledge Graph

In the following, we detail the finalized knowledge graph utilized in our implementation. Table 5 summarizes the semantic mappings $\mathcal{M}_{c\rightarrow a}$ and $\mathcal{M}_{c\rightarrow l}$. The semantic mappings $\mathcal{M}_{a\rightarrow c}$ and $\mathcal{M}_{a\rightarrow l}$ are represented in Table 6. Finally, the semantic mappings $\mathcal{M}_{l\rightarrow c}$ and $\mathcal{M}_{l\rightarrow a}$ are represented in Table 7.

### B.2  Impact of the Knowledge Graph Size on KRPS

In this section, we systematically evaluate the influence of knowledge graph completeness on the performance of KRPS. Motivated by the necessity to ascertain how the granularity and extent of semantic data affect predictive outcomes, the completeness of the knowledge graph is varied at

| Label Name | Abbreviation |
|---|---|
| Car | Car |
| Medium Vehicle | MedVeh |
| Large Vehicle | LarVeh |
| Bus | Bus |
| Motorbike | Mobike |
| Emergency Vehicle | EmVeh |
| Pedestrian | Ped |
| Cyclist | Cyc |
| Vehicle Traffic Light | TL |
| Other Traffic Light | OthTL |

Table 2: List of agent classes and their abbreviations as reported in the ROAD dataset [1].

| Label | Abbreviation |
|---|---|
| In vehicle lane | VehLane |
| In outgoing lane | OutgoLane |
| In incoming lane | IncomLane |
| In outgoing cycle lane | OutgoCycLane |
| In incoming cycle lane | IncomCycLane |
| On left pavement | LftPav |
| On right pavement | RhtPav |
| On pavement | Pav |
| At junction | Jun |
| At crossing | Xing |
| At bus stop | BusStop |
| At parking | parking |

Table 3: List of used location classes and their abbreviations as reported in the ROAD dataset [1].

| Label | Abbreviation |
|---|---|
| Moving away | MovAway |
| Moving towards | MovTow |
| Moving | Mov |
| Braking | Brake |
| Stopped | Stop |
| Indicating left | IncatLft |
| Indicating right | IncatRht |
| Hazard lights on | HazLit |
| Turning left | TurnLft |
| Turning right | TurnRht |
| Moving right | MovRht |
| Moving left | MovLft |
| Overtaking | Ovtak |
| Waiting to cross | Wait2X |
| Crossing road from left | XingFmLft |
| Crossing road from right | XingFmRht |
| Crossing | Xing |
| Pushing object | PushObj |
| Traffic light red | Red |
| Traffic light amber | Amber |
| Traffic light green | Green |

Table 4: List of used action classes and their abbreviations as reported in the ROAD dataset [1]

| Agent | List of Actions | List of Locations |
|---|---|---|
| Ped | MovAway,MovTow,Mov,Stop,Wait2X,XingFmLft,XingFmRht, | VehLane,IncomLane,Pav,LftPav,RhtPav,Jun,xing,BusStop |
| Car | MovAway,MovTow,Brake,Stop,IncatLft,IncatRht,TurLft,TurRht | VehLane,OutgoLane,IncomLane,Jun |
| Cyc | MovAway,MovTow,Stop,TurLft,XingFmLft | VehLane,OutgoLane,OutgoCycLane,IncomLane,IncomCycLane |
| Mobike | MovAway,MovTow,Brake,Stop,IncatLft,IncatRht,TurLft,TurRht | VehLane,OutgoLane,IncomLane,Jun |
| MedVeh | MovTow,Stop,TurRht, TurLft, Brake | IncomLane,Jun,OutgoLane |
| LarVeh | MovTow,Stop,TurRht, TurLft, Brake | IncomLane,Jun,OutgoLane |
| Bus | MovTow,Stop,XingFmLft | VehLane,IncomLane,Jun, BusStop |

Table 5: The semantic mappings between the agent classes and the action classes ($\mathcal{M}_{c \to a}$), and the agent classes and the location classes ($\mathcal{M}_{c \to l}$).

| Action | List of Agents | List of Locations |
|---|---|---|
| MovAway | Ped,Car,Cyc,MedVeh,Bus,LarVeh | VehLane,OutgoLane,OutgoCycLane,Pav,LftPav,RhtPav,Jun |
| MovTow | Ped,Car,Cyc,MedVeh,Bus,LarVeh | VehLane,IncomLane,IncomCycLane,LftPav,RhtPav,Jun |
| Mov | Ped | Pav |
| Brake | Car | VehLane,Jun |
| Stop | Ped,Car,Cyc,MedVeh,Bus | VehLane,IncomLane,IncomCycLane,Pav,LftPav,RhtPav,Jun,BusStop |
| IncatLft | Car | VehLane,Jun |
| IncatRht | Car | IncomLane,Jun |
| TurLft | Car,Cyc | VehLane,Jun |
| TurRht | Car,MedVeh | IncomLane,Jun |
| Ovtak | Car | VehLane |
| Wait2X | Ped | LftPav,RhtPav |
| XingFmLft | Ped,Car,Cyc,Bus | VehLane,IncomLane,Jun,xing |
| XingFmRht | Ped | VehLane,IncomLane,RhtPav,Jun |
| Xing | Ped,Cyc | Xing |
| PushObj | Ped | LftPav,RhtPav |

Table 6: The semantic mappings between the action classes and the agent classes ($\mathcal{M}_{a \to c}$), and the agent classes and location classes ($\mathcal{M}_{a \to l}$).

| Location | List of Agents | List of Actions |
|---|---|---|
| VehLane | Ped,Car,Cyc,Bus | MovAway,MovTow,Brake,Stop,IncatLft,TurLft,XingFmLft,XingFmRht |
| OutgoLane | Car,Cyc | MovAway |
| OutgoCycLane | Cyc | MovAway |
| IncomLane | Ped,Car,Cyc,MedVeh,Bus | MovTow,Stop,IncatRht,TurRht,XingFmLft,XingFmRht |
| IncomCycLane | Cyc | MovTow,Stop |
| Pav | Ped | MovAway,Mov,Stop |
| LftPav | Ped,Cyc | MovAway,MovTow,Stop,Wait2X,PushObj |
| RhtPav | Ped | MovAway,MovTow,Stop,Wait2X,XingFmRht,PushObj |
| Jun | Ped,Car,Cyc,MedVeh,Bus | MovAway,MovTow,Brake,Stop,IncatLft,IncatRht,TurLft,TurRht,XingFmLft,XingFmRht |
| xing | Ped | XingFmLft |
| BusStop | Ped,Bus | Stop |
| parking | Car | parking |

Table 7: The semantic mappings between the location classes and the agent classes ($\mathcal{M}_{l \to c}$), and the location classes and action classes ($\mathcal{M}_{l \to a}$).

50%, 70%, and 90% of its full capacity. The Results in Table 8 report the results on the ROAD dataset with $\alpha = 0.1$ for the $\{agent \to location\}$ sequence, using the "$1 - softmax$" score with KRPS. The results reveal a decrease in semantic consistency corresponding with reduced knowledge graph completeness, although statistical coverage guarantees are maintained. Furthermore the set size is larger with reduced portions of the knowledge graph. This result is expected, as the removed connections from the reduced knowledge graph cannot be used during the refinement process, which impacts the set size and the semantic consistency w.r.t to the full graph. It's important to note that while semantic consistency decreases, Theorem 3.1 and Corollary 3.1 remain valid. This is because these theorems are formulated with respect to the currently used portion of the knowledge graph; therefore, semantic consistency remains perfect (value of 1) when considered within the context of the available graph segment.

| Metrics | % of used knowledge graph | | | |
|---|---|---|---|---|
| | 50% | 70% | 90% | 100% |
| DTC | 0.07 | 0.06 | 0.05 | **0.02** |
| SS | 6.82 | 6.53 | 5.96 | **5.86** |
| SC (w.r.t full K) | 0.83 | 0.86 | 0.98 | **1.00** |
| SC (w.r.t reduced K) | 1.00 | 1.00 | 1.00 | **1.00** |

Table 8: Effects of the knowledge graph size on the performance of KRPS. We evaluate the performance of KRPS with different sizes of the knowledge graph (50%, 70%, 90%) and compare it to the reference of 100% of the knowledge graph (designated in **Bold** in the table). The results are obtained for $\alpha = 0.1$ for the $\{agent \to location\}$ sequence, using the "$1 - softmax$" score with KRPS.

| Task | Score | Method | α = 0.1 | | | α = 0.2 | | | α = 0.3 | | | α = 0.4 | | | α = 0.5 | | |
|---|---|---|---|---|---|---|---|---|---|---|---|---|---|---|---|---|---|
| | | | SS | DTC | SC | SS | DTC | SC | SS | DTC | SC | SS | DTC | SC | SS | DTC | SC |
| Location | APS | Stand | 7.12 | 0.05 | 0.80 | 6.07 | 0.10 | 0.79 | 5.58 | 0.16 | 0.69 | 4.61 | 0.20 | 0.76 | 4.20 | 0.27 | 0.61 |
| | | KRPS | 5.90 | 0.02 | 1.00 | 4.96 | 0.05 | 1.00 | 4.05 | 0.01 | 1.00 | 3.65 | 0.12 | 1.00 | 2.61 | 0.00 | 1.00 |
| | RAPS | Stand | 2.66 | 0.07 | 0.89 | 2.03 | 0.15 | 0.93 | 1.70 | 0.25 | 0.95 | 1.56 | 0.34 | 0.96 | 1.46 | 0.43 | 0.95 |
| | | KRPS | 2.18 | 0.05 | 1.00 | 1.78 | 0.13 | 1.00 | 1.58 | 0.14 | 1.00 | 1.45 | 0.13 | 1.00 | 1.36 | 0.22 | 1.00 |
| Action | APS | Stand | 9.77 | 0.05 | 0.75 | 8.01 | 0.10 | 0.73 | 8.14 | 0.20 | 0.72 | 5.72 | 0.30 | 0.70 | 6.56 | 0.14 | 0.69 |
| | | KRPS | 7.29 | 0.02 | 1.00 | 6.01 | 0.03 | 1.00 | 5.59 | 0.01 | 1.00 | 4.35 | 0.11 | 1.00 | 3.94 | 0.00 | 1.00 |
| | RAPS | Stand | 4.67 | 0.07 | 0.84 | 3.94 | 0.13 | 0.87 | 3.99 | 0.23 | 0.86 | 2.54 | 0.25 | 0.92 | 2.95 | 0.37 | 0.91 |
| | | KRPS | 3.82 | 0.03 | 1.00 | 3.26 | 0.02 | 1.00 | 3.27 | 0.13 | 1.00 | 2.22 | 0.15 | 1.00 | 2.52 | 0.17 | 1.00 |

Table 9: Results on the ROAD dataset for the task sequences $\{agent \rightarrow location\}$ and $\{agent \rightarrow action\}$.

| Task | Score | Method | α = 0.1 | | | α = 0.2 | | | α = 0.3 | | | α = 0.4 | | | α = 0.5 | | |
|---|---|---|---|---|---|---|---|---|---|---|---|---|---|---|---|---|---|
| | | | SS | DTC | SC | SS | DTC | SC | SS | DTC | SC | SS | DTC | SC | SS | DTC | SC |
| Location | APS | Stand | 9.02 | 0.07 | 0.79 | 7.92 | 0.10 | 0.76 | 5.97 | 0.18 | 0.73 | 4.03 | 0.22 | 0.71 | 3.98 | 0.29 | 0.64 |
| | | KRPS | 7.50 | 0.01 | 1.00 | 5.99 | 0.06 | 1.00 | 4.00 | 0.01 | 1.00 | 3.24 | 0.14 | 1.00 | 2.73 | 0.05 | 1.00 |
| | RAPS | Stand | 2.45 | 0.09 | 0.87 | 2.07 | 0.15 | 0.91 | 1.87 | 0.23 | 0.93 | 1.83 | 0.34 | 0.96 | 1.40 | 0.40 | 0.95 |
| | | KRPS | 2.01 | 0.04 | 1.00 | 2.04 | 0.10 | 1.00 | 1.35 | 0.14 | 1.00 | 1.17 | 0.09 | 1.00 | 1.10 | 0.09 | 1.00 |
| Action | APS | Stand | 10.41 | 0.05 | 0.77 | 6.84 | 0.13 | 0.74 | 6.02 | 0.15 | 0.74 | 5.36 | 0.12 | 0.78 | 4.83 | 0.20 | 0.68 |
| | | KRPS | 5.98 | 0.02 | 1.00 | 4.18 | 0.02 | 1.00 | 3.90 | 0.07 | 1.00 | 3.87 | 0.12 | 1.00 | 3.94 | 0.08 | 1.00 |
| | RAPS | Stand | 4.36 | 0.06 | 0.87 | 3.03 | 0.15 | 0.89 | 2.48 | 0.16 | 0.86 | 1.50 | 0.24 | 0.94 | 1.20 | 0.37 | 0.90 |
| | | KRPS | 3.57 | 0.04 | 1.00 | 2.88 | 0.02 | 1.00 | 2.03 | 0.09 | 1.00 | 1.07 | 0.07 | 1.00 | 1.02 | 0.10 | 1.00 |

Table 10: Results on the Waymo/ROAD++ dataset for the task sequences $\{agent \rightarrow location\}$ and $\{agent \rightarrow action\}$.

# C  Results for Further Task Sequences

In Section 4.5, we reported results for sequences of 2 tasks: $\{agent \rightarrow location\}$ and $\{agent \rightarrow action\}$ for $\alpha = [0.1, 0.2, 0.4]$. In the following, we present more results on both datasets for the full set of values of $\alpha = [0.1, 0.2, 0.3, 0.4, 0.5]$ for the sequences $\{agent \rightarrow location\}$ and $\{agent \rightarrow action\}$ in Table 9 and Table 10, respectively. Furthermore, we present results using sequences of 3 tasks on the ROAD dataset to show the capability of KRPS to handle sequences with higher numbers of tasks and in different task orders. The 3-tasks sequences that we consider are: $Seq1 : \{agent \rightarrow action \rightarrow location\}$, and $Seq2 : \{location \rightarrow action \rightarrow agent\}$. We use the same evaluation set-up and data splits reported in the evaluation section in our paper. The results are reported in Table11 and Table 12, respectively.

In all task sequences, KRPS still holds the theoretical coverage guarantees for all values of $\alpha$. More importantly, KRPS achieves the desired coverage rates while being able to reduce the set size considerably. The conditional semantic consistency also holds, as it is guaranteed by Theorem 3.1 and Corollary 3.1. For all task sequences, the conditional semantic consistency for task 2 with respect to task 1 and $\mathcal{K}$, task 3 with respect to task 2 and $\mathcal{K}$, and task 3 with respect to task 1 and $\mathcal{K}$, is 100%.

# D  Qualitative Results

In this section, we present further qualitative results on the ROAD and Waymo/ROAD++ datasets for the task sequences $\{agent \rightarrow action\}$, $\{agent \rightarrow location\}$, and $\{agent \rightarrow action \rightarrow location\}$.

| Task | Score | Method | α = 0.1 | | | α = 0.2 | | | α = 0.3 | | | α = 0.4 | | | α = 0.5 | | |
|---|---|---|---|---|---|---|---|---|---|---|---|---|---|---|---|---|---|
| | | | SS | DTC | SC | SS | DTC | SC | SS | DTC | SC | SS | DTC | SC | SS | DTC | SC |
| Action | APS | Stand | 9.82 | 0.05 | 0.71 | 8.14 | 0.10 | 0.65 | 6.91 | 0.15 | 0.61 | 5.87 | 0.21 | 0.58 | 5.10 | 0.26 | 0.56 |
| | | KRPS | 7.12 | 0.01 | 1.00 | 5.84 | 0.02 | 1.00 | 4.79 | 0.00 | 1.00 | 3.88 | 0.01 | 1.00 | 3.17 | 0.00 | 1.00 |
| | RAPS | Stand | 4.73 | 0.07 | 0.84 | 3.98 | 0.13 | 0.87 | 3.16 | 0.18 | 0.9 | 2.61 | 0.26 | 0.92 | 2.20 | 0.33 | 0.93 |
| | | KRPS | 3.84 | 0.06 | 1.00 | 3.26 | 0.12 | 1.00 | 2.65 | 0.17 | 1.00 | 2.24 | 0.15 | 1.00 | 1.93 | 0.21 | 1.00 |
| Location | APS | Stand | 7.67 | 0.05 | 0.83 | 6.77 | 0.14 | 0.82 | 6.19 | 0.21 | 0.80 | 5.80 | 0.22 | 0.78 | 5.31 | 0.24 | 0.77 |
| | | KRPS | 6.33 | 0.01 | 1.00 | 5.30 | 0.00 | 1.00 | 4.53 | 0.08 | 1.00 | 3.86 | 0.00 | 1.00 | 3.23 | 0.01 | 1.00 |
| | RAPS | Stand | 3.03 | 0.07 | 0.90 | 2.38 | 0.16 | 0.94 | 2.04 | 0.25 | 0.95 | 1.83 | 0.35 | 0.96 | 1.69 | 0.35 | 0.96 |
| | | KRPS | 2.06 | 0.07 | 1.00 | 2.12 | 0.16 | 1.00 | 1.80 | 0.04 | 1.00 | 1.69 | 0.13 | 1.00 | 1.57 | 0.13 | 1.00 |

Table 11: Results on the ROAD dataset for the task sequence $\{agent \rightarrow action \rightarrow location\}$.

| Task | Score | Method | $\alpha = 0.1$ | | | $\alpha = 0.2$ | | | $\alpha = 0.3$ | | | $\alpha = 0.4$ | | | $\alpha = 0.5$ | | |
|---|---|---|---|---|---|---|---|---|---|---|---|---|---|---|---|---|---|
| | | | SS | DTC | SC | SS | DTC | SC | SS | DTC | SC | SS | DTC | SC | SS | DTC | SC |
| Action | APS | Stand | 9.82 | 0.05 | 0.72 | 8.15 | 0.00 | 0.68 | 6.91 | 0.15 | 0.64 | 5.87 | 0.21 | 0.60 | 5.10 | 0.26 | 0.58 |
| | | KRPS | 7.35 | 0.00 | 1.00 | 6.02 | 0.00 | 1.00 | 4.91 | 0.00 | 1.00 | 3.97 | 0.00 | 1.00 | 3.20 | 0.01 | 1.00 |
| | RAPS | Stand | 4.73 | 0.07 | 0.82 | 3.98 | 0.15 | 0.85 | 3.16 | 0.20 | 0.89 | 2.61 | 0.26 | 0.91 | 2.20 | 0.35 | 0.92 |
| | | KRPS | 3.86 | 0.06 | 1.00 | 3.24 | 0.13 | 1.00 | 2.64 | 0.18 | 1.00 | 2.23 | 0.25 | 1.00 | 1.93 | 0.32 | 1.00 |
| Agent | APS | Stand | 7.00 | 0.06 | 0.53 | 6.52 | 0.13 | 0.54 | 6.16 | 0.20 | 0.55 | 5.76 | 0.27 | 0.56 | 5.48 | 0.33 | 0.55 |
| | | KRPS | 3.45 | 0.00 | 1.00 | 3.07 | 0.00 | 1.00 | 2.74 | 0.00 | 1.00 | 2.36 | 0.00 | 1.00 | 2.03 | 0.01 | 1.00 |
| | RAPS | Stand | 1.55 | 0.08 | 0.92 | 1.31 | 0.08 | 0.92 | 1.22 | 0.27 | 0.97 | 1.16 | 0.27 | 0.97 | 1.13 | 0.26 | 0.95 |
| | | KRPS | 1.31 | 0.08 | 1.00 | 1.19 | 0.10 | 1.00 | 1.14 | 0.17 | 1.00 | 1.11 | 0.27 | 1.00 | 1.08 | 0.26 | 1.00 |

Table 12: Results on the ROAD dataset for the task sequence $\{location \rightarrow action \rightarrow agent\}$.

## D.1 Qualitative Results on the ROAD Dataset

Figure 4 shows 3 different scenes from the ROAD dataset with the agent of interest highlighted with the red bounding box. For each bounding box, we perform the indicated CP procedure to acquire the prediction sets for the agent classification task. Based on the agent prediction sets, we report the generated prediction sets with and without KRPS using the Softmax, APS, and RAPS scores.

Figure 6a shows a scene with a *car stopping in the outgoing lane*. The prediction sets for the action and location classification tasks demonstrate how KRPS achieved a substantial reduction for the prediction sets. This reduction is particularly observable for the location task with the softmax and APS scores, where the prediction set size is reduced by 4 classes.

Figure 4b shows a scene with a *bus stopping in the vehicle lane*. All approaches succeed in including the correct labels in the prediction sets. The combination of KRPS with RAPS succeeds in constructing a singleton for the tasks location and action classification. By applying KRPS, the action and location classes that are not relevant to the agent class are removed.

Figure 6b showcases a scene with a *bus moving towards in the incoming lane*. Using KRPS, the size of the prediction sets for the action classification is reduced by $50\%$ for the softmax score, by $75\%$ for APS, and by $66\%$ for RAPS. For the location classification task, KRPS reduced the prediction set size by $50\%$ for the softmax score, by $50\%$ for APS. For RAPS, the prediction set is not reduced, since both locations, *incoming lane* and *junction* are possible, given the agent class and its action.

The figures highlight the capability of KRPS to reduce the set size by removing action and location classes that are not relevant to the agent. Theorem 3.1 ensures that this removal procedure does not affect the marginal coverage property.

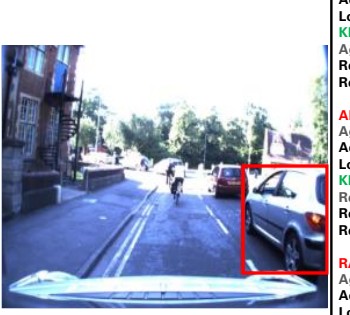

**Softmax**
Agent(5): ['Car' 'MedVeh' 'EmVeh' 'LarVeh' 'Bus']
Action(4): ['Stop' 'IncatLft' 'Brake' 'IncatRht']
Location(7): ['IncomLane' 'OutgoLane' 'Jun' 'RhtPav' 'IncomCycLane' 'parking' 'OutgoCycLane']
KRPS :
Agent(5): ['Car' 'MedVeh' 'EmVeh' 'LarVeh' 'Bus']
Refined Action(4): ['Stop' 'IncatLft' 'Brake' 'IncatRht']
Refined Location(3): ['IncomLane' 'OutgoLane' 'Jun']

**APS**
Agent(5): ['Car' 'MedVeh' 'EmVeh' 'LarVeh' 'Bus']
Action(6): ['Stop' 'IncatLft' 'Brake' 'IncatRht' 'HazLit' 'MovAway']
Location(7): ['IncomLane' 'OutgoLane' 'Jun' 'RhtPav' 'IncomCycLane' 'parking''OutgoCycLane']
KRPS :
Refined Agent(5): ['Car' 'MedVeh' 'EmVeh' 'LarVeh' 'Bus']
Refined Action(5):['Stop' 'IncatLft' 'Brake' 'IncatRht' 'MovAway']
Refined Location(3): ['IncomLane' 'OutgoLane' 'Jun']

**RAPS**
Agent(1):['Car']
Action(2): ['Stop' 'IncatLft']
Location(4): ['IncomLane' 'OutgoLane' 'Jun' 'RhtPav']
KRPS :
Agent(1): ['Car']
Refined Action(2): ['Stop' 'IncatLft']
Refined Location(3): ['IncomLane' 'OutgoLane' 'Jun']

(a) Scene of a *car stopping in the outgoing lane* from the ROAD dataset [1] with prediction sets using 3 scoring functions for CP (Softmax, APS, RAPS) without and with KRPS.

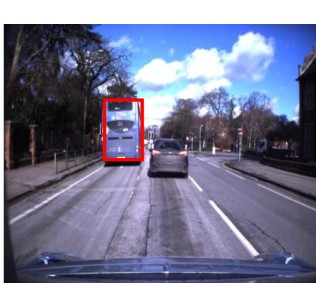

**Softmax**
Agent(8): ['Bus' 'MedVeh' 'LarVeh' 'Cyc' 'Mobike' 'EmVeh' 'TL' 'OthTL']
Action(7): ['MovAway' 'Stop' 'Brake' 'HazLit' 'IncatRht' 'IncatLft' 'Amber']
Location(2): ['VehLane' 'OutgoCycLane']
KRPS :
Agent(8): ['Bus' 'MedVeh' 'LarVeh' 'Cyc' 'Mobike' 'EmVeh' 'TL' 'OthTL']
Refined Action(2): ['MovAway' 'Stop']
Refined Location(2): ['VehLane' 'OutgoCycLane']

**APS**
Agent(8): ['Bus' 'MedVeh' 'LarVeh' 'Cyc' 'Mobike' 'EmVeh' 'TL' 'OthTL']
Action(8): ['MovAway' 'Stop' 'Brake' 'HazLit' 'IncatRht' 'IncatLft' 'Amber' 'Xing''Ovtak']
Location(2): ['VehLane' 'OutgoCycLane']
KRPS :
Agent(8): ['Bus' 'MedVeh' 'LarVeh' 'Cyc' 'Mobike' 'EmVeh' 'TL' 'OthTL']
Refined Action(2):['MovAway' 'Stop']
Refined Location(2): ['VehLane' 'OutgoCycLane']

**RAPS**
Agent(1):['Bus']
Action(2): ['MovAway' 'Stop' 'Brake' 'HazLit']
Location(1): ['VehLane']
KRPS :
Agent(1): ['Bus']
Refined Action(1): ['Stop']
Refined Location(1): ['VehLane']

(b) Scene of a *bus stopping in the vehicle lane* from the ROAD dataset [1] with prediction sets using 3 scoring functions for CP (Softmax, APS, RAPS) without and with KRPS.

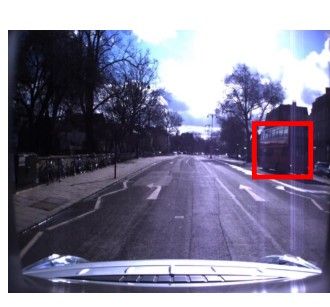

**Softmax**
Agent(7): ['Bus' 'LarVeh' 'EmVeh' 'MedVeh' 'TL' 'Ped' 'OthTL']
Action(10): ['MovTow' 'IncatLft' 'Stop' 'IncatRht' 'TurLft' 'HazLit' 'Ovtak' 'TurRht' 'XingFmLft' 'XingFmRht']
Location(8): ['IncomLane' 'Jun' 'IncomCycLane' 'OutgoLane' 'parking' 'OutgoCycLane' 'VehLane' 'BusStop']
KRPS:
Agent(7): ['Bus' 'LarVeh' 'EmVeh' 'MedVeh' 'TL' 'Ped' 'OthTL']
Refined Action(5): ['MovTow' 'Stop' 'TurRht' 'XingFmLft' 'XingFmRht']
Refined Location(4): ['IncomLane' 'Jun' 'VehLane' 'BusStop']

**APS**
Agent(7): ['Bus' 'LarVeh' 'EmVeh' 'MedVeh' 'TL' 'Ped' 'OthTL']
Action(10): ['MovTow' 'IncatLft' 'Stop' 'IncatRht' 'TurLft' 'HazLit' 'Ovtak' 'TurRht''XingFmLft' 'XingFmRht']
Location(8): ['IncomLane' 'Jun' 'IncomCycLane' 'OutgoLane' 'parking' 'OutgoCycLane''VehLane' 'BusStop' ]
KRPS:
Agent(7): ['Bus' 'LarVeh' 'EmVeh' 'MedVeh' 'TL' 'Ped' 'OthTL']
Refined Action(6):['MovTow' 'Stop' 'TurRht' 'XingFmLft' 'XingFmRht' 'Wait2X']
Refined Location(4): ['IncomLane' 'Jun' 'VehLane' 'BusStop']

**RAPS**
Agent(1):['Bus']
Action(6): ['MovTow' 'IncatLft' 'Stop' 'IncatRht' 'TurLft' 'HazLit']
Location(2): ['IncomLane' 'Jun']
KRPS:
Agent(1): ['Bus']
Refined Action(2): ['MovTow' 'Stop']
Refined Location(2): ['IncomLane' 'Jun']

(c) Scene of a *bus moving towards in the incoming lane* from the ROAD dataset [1] with prediction sets using 3 scoring functions for CP (Softmax, APS, RAPS) without and with KRPS.

Figure 4: Scenes from the ROAD dataset [1] with prediction sets using 3 scoring functions for CP (Softmax, APS, RAPS) without and with KRPS.

### D.2   Qualitative Results on the Waymo/ROAD++ Dataset

Figure 5 shows 2 different scenes from the Waymo/ROAD++ dataset, characterized by challenging situational and environmental conditions that may induce high uncertainty. The agent of interest is highlighted with the red bounding box. We perform the indicated CP procedure for each bounding box to acquire the prediction sets for the agent classification task. Based on the agent prediction sets, we report the generated prediction sets with and without KRPS using the Softmax, APS, and RAPS scores.

Figure 5a depicts a low-light scenario where a pedestrian crosses the street with a car in the background of the bounding box. The challenging lighting conditions and complex scene composition contribute to uncertainty, prompting the model to assign vehicle-associated actions and locations such as *Brake* and *outgoing lane*. using KRPS mitigates this confusion by restricting the subsequent tasks to consider only classes suitable for pedestrians or bicycles, as determined by the agent classification model. KRPS notably reduces uncertainty and shrinks the prediction set size by 80%, 83%, and 50% for softmax, APS, and RAPS predictions, respectively, for the action classification task. For location classification, the prediction set size is reduced by 50%, 75%, and 50% for softmax, APS, and RAPS, respectively.

Figure 5b illustrates a scenario where a pedestrian is crossing the street while pushing a bicycle. This presence of the bicycle often misleads models for action and location to assign characteristics typical of bicyclists, such as *Brake* and *outgoing lane*. KRPS addresses this issue by ensuring that only classes suitable for either pedestrians or bicycles are considered, as dictated by the initial agent classification results. This application of KRPS significantly reduces uncertainty and narrows the prediction set size by 71%, 60%, and 66% for the softmax, APS, and RAPS predictions for the action classification task, respectively. Similarly, for the location classification task, the prediction set sizes are reduced by 66%, 50%, and 66% for softmax, APS, and RAPS, respectively.

## E   Further Details About Empty Prediction Sets

In the KRPS framework, the occurrence of empty prediction sets is a critical aspect, designed to signal either anomalies in the detection or significant mismatches and gaps within the knowledge graph. The empty sets arise when no available classes meet the semantic and statistical criteria established by the knowledge graph and conformal prediction rules. In traditional conformal prediction, an empty prediction set is generated when the non-conformity score for any potential prediction exceeds the calibrated threshold, indicating that none of the possible outcomes is statistically plausible within the given confidence level. KRPS enhances the traditional conformal prediction model by integrating a semantic layer that evaluates the coherence of potential predictions with the knowledge graph. This integration means that even if a prediction is statistically plausible, it might still be rejected if it fails to align semantically with the knowledge graph. Hence, empty prediction sets may also occur due to the absence of semantically consistent labels, particularly in scenarios where the knowledge graph does not fully cover all possible real-world variations.

We conduct an experiment to quantify the frequency of empty prediction sets produced by KRPS and compare these outcomes to traditional conformal prediction baselines, to assess the impact of semantic layer in KRPS on the generation of empty sets. The results, depicted in Figure 6, show a trend where $1 - Softmax$, APS, and their combination with KRPS produce more empty sets as the value of $\alpha$ increases, a behavior expected as higher $\alpha$ values lead conformal prediction approaches to generate smaller prediction sets, as they tolerate more risk. This is however different for RAPS, where the percentage of empty sets is zero across the different values of $\alpha$. This is an expected behavior since the regularization term in RAPS encourages the inclusion of additional labels if the set is too small, thus preventing it from being empty. By adaptively adjusting the threshold for inclusion based on the desired coverage level, RAPS ensures that there is always at least one label that meets the criteria, thereby avoiding empty prediction sets. KRPS, in contrast to the baselines, consistently shows a higher frequency of empty sets due to the semantic consistency checks integrated. If a po-

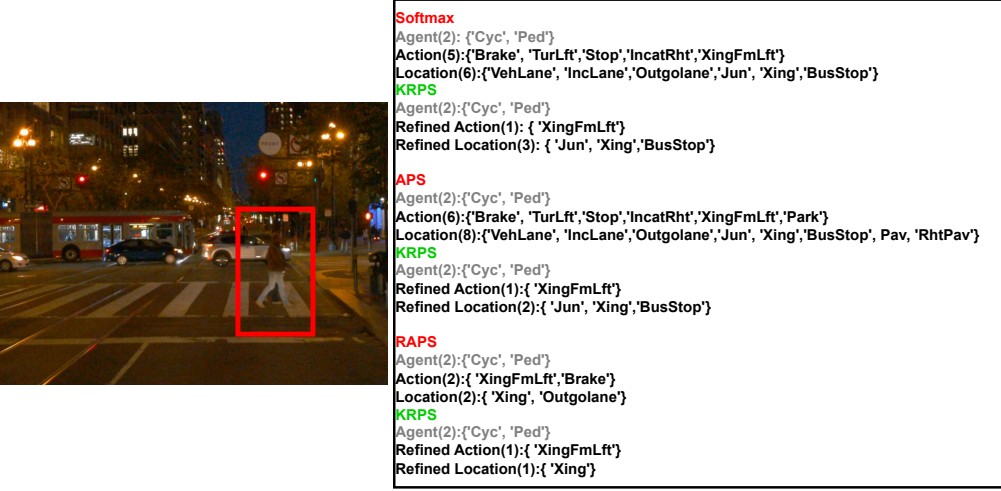

(a) Scene of a *pedestrian pushing a bicycle and crossing the street* from the Waymo/ROAD++ dataset with prediction sets using 3 scoring functions for CP (Softmax, APS, RAPS) without and with KRPS.

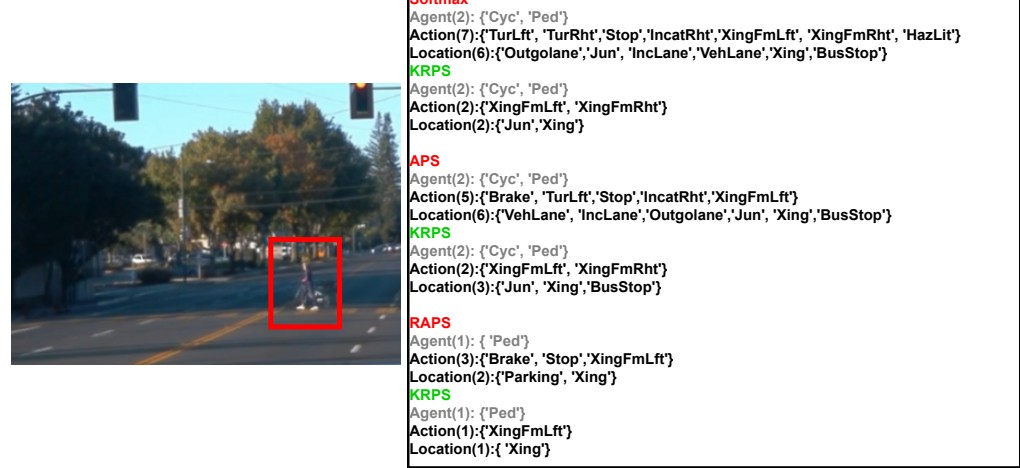

(b) Scene of a *pedestrian pushing a bicycle and crossing the street* from the Waymo/ROAD++ dataset with prediction sets using 3 scoring functions for CP (Softmax, APS, RAPS) without and with KRPS.

Figure 5: Scenes from the Waymo/ROAD++ dataset with prediction sets using 3 scoring functions for CP (Softmax, APS, RAPS) without and with KRPS.

tential prediction fails to meet both semantic and statistical criteria, KRPS opts to return an empty set, avoiding inaccurate or contextually inappropriate predictions. Conversely, traditional conformal prediction methods, which lack a semantic filter, tend to produce fewer empty sets but at the risk of including predictions that, although statistically plausible, might not align semantically with the context of the scene.

In Figure 7, we show a qualitative example of a situation where KRPS outputs an empty set, while the RAPS does not. The scenario illustrates a car for which the ground truth location is *VehLane*, which indicates a *Vehicle lane*. Applying RAPS using $\alpha = 0.5$ leads to a prediction set including a single class, which is *OutgoCycLane*, referring to an *Outgoing Cycle Lane*. In contrast, when refining the prediction set with KRPS in accordance with the agent class *Car* and the knowledge graph, KRPS yields an empty set. This outcome signals that none of the predicted locations are applicable to the *Car* agent class, prompting considerations of potential anomalies, such as a car mistakenly detected in a cycle lane. It is important to note that had such scenarios been present

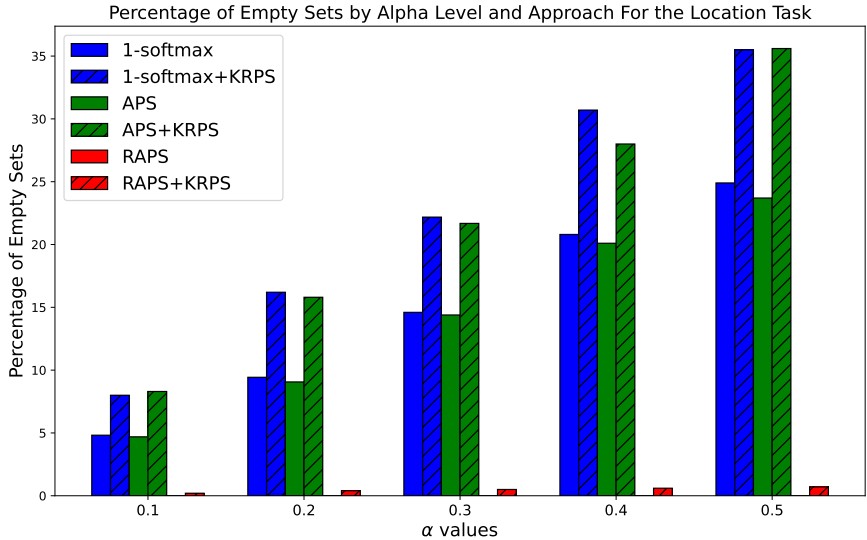

(a) Percentage of empty sets for different values of $\alpha$ for the location classification task.

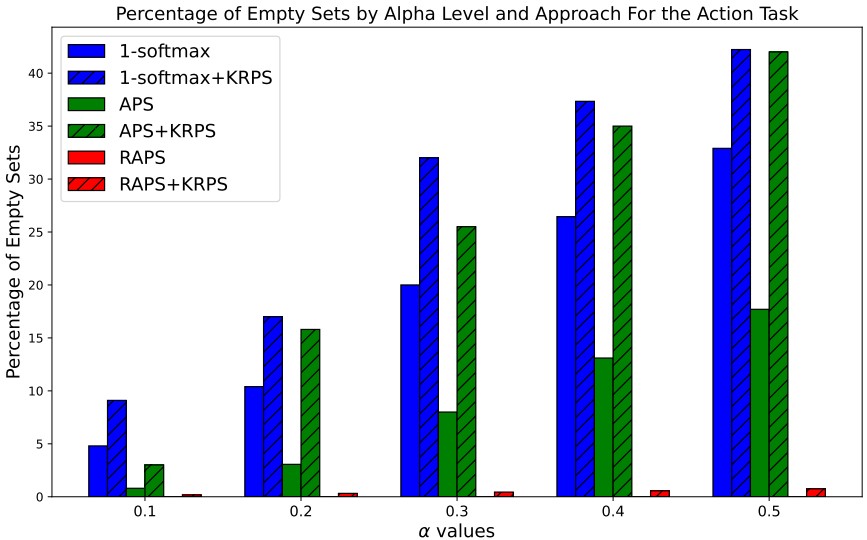

(b) Percentage of empty sets for different values of $\alpha$ for the action classification task.

Figure 6: Percentage of empty sets for different values of $\alpha$ for the location and action classification tasks based w.r.t the knowledge graph and predictions of the agent class.

in the training data, the knowledge graph would have accounted for them, preventing KRPS from producing an empty set under these conditions.

## F More on the Practicality of KRPS

While our paper focuses on three high-level perception tasks for urban scene understanding, KRPS can be extended to other robot perception applications with varying natures and a number of tasks. The formulation of Algorithm 1 and Theorem 3.1 is adaptable to diverse environmental contexts.

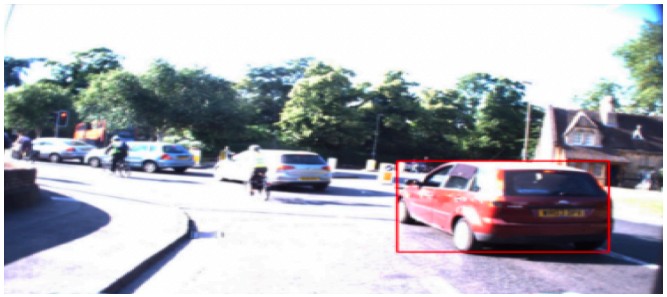

Detected agent class: {'Car'}
RAPS Location: {OutgoCycLane}
KRPS: {}

Figure 7: Qualitative example of empty set generated by KRPS, using RAPS and $\alpha = 0.5$.

For example, KRPS can be applied to urban search and rescue, where tasks such as victim detection, hazard identification, and robot action selection are critical [53]. In this scenario, the detection of victims informs subsequent hazard identification tasks by highlighting areas where hazards are likely to be present. The results from hazard identification can then guide the action selection of the robot, such as navigating safely through debris or delivering emergency supplies. The role of external knowledge in the form of knowledge graphs or other forms of semantics can be decisive for the mission success [54]. KRPS enhances this application by ensuring that each prediction set is semantically consistent with the preceding tasks and by providing uncertainty quantification, which is crucial for making reliable decisions in high-stakes environments.

In the domain of robot manipulation, KRPS can significantly improve the accuracy and reliability of tasks such as object recognition, grasp strategy selection, and precise placement. Here, the classification of objects directly influences the choice of the manipulation action [55]. For instance, identifying an object as fragile necessitates a more delicate grasping approach compared to a robust item. Uncertainty quantification is particularly important in this context to avoid mistakes such as misidentifying an object, which could lead to improper handling and potential damage. KRPS ensures that each stage of the manipulation process is informed by the previous task's outcomes, enhancing both semantic consistency and decision reliability.

Additionally, KRPS can be utilized in robotic navigation, especially in dynamic and unpredictable urban environments. The class of the robot's behavior, such as stopping, turning, or accelerating, can depend on the classes of agents present in the environment and their predicted future maneuvers [56, 57]. For example, the presence of a pedestrian crossing the road might necessitate the robot to stop, while an approaching vehicle could require the robot to yield or change lanes [51, 52]. In this application, KRPS provides a structured way to integrate these interdependent tasks, ensuring that navigation decisions are not only statistically valid but also semantically coherent with the surrounding context. This integration of uncertainty quantification and semantic consistency enables robots to navigate more safely and efficiently, adapting to complex real-world scenarios with greater reliability.

