# OpenReview forum: "Conformal Prediction for Semantically-Aware Autonomous Perception in Urban Environments"
_robot-learning.org/CoRL/2024/Conference — CoRL 2024_

### Official Review · Reviewer_UhpA · 2024-07-14

**Originality:** 3
**Technical Quality:** 3
**Clarity Of Presentation:** 3
**Potential Impact:** 2
**Recommendation:** 3
**Confidence:** 4

**Review:**

Overall, the paper is well written. Conformal prediction is
an important and promising direction for high-dimensional
settings such as perception. The neuro-symbolic approach
seems like a natural extension of standard conformal
prediction. I also appreciate the evaluation over two
different case studies. I do have a few comments that
can be used to improve the paper, however.

My first comment has to do with the knowledge graph
construction. The authors do list this as one of the
method's limitations but it is not clear how well such
graphs can be constructed from high-dimensional data.
The authors at one point mention that implications can
be developed from common sense but such a manual
approach is unlikely to scale and generalize beyond
very simple scenarios.

Related to the above, the proposed method may
output an empty prediction set but it is not clear
how often that happens and whether it may be an issue
in practice. The authors are encouraged to elaborate on
this issue since standard conformal prediction is
designed to be conservative -- thus, the comparison
may be unfair.

One more related comment is that the correction
procedure is not explained anywhere in the main body
or in the appendix. What does this correction entail?
Presumably the first task may need to include more
labels in the predicted set so as to ensure conformity
over all tasks? It is essential to include this in
the paper, otherwise it would be incomplete.

Technical:

- The authors mention in a few places that the DTC
  metric must be positive but I am not sure if this
  is correct. The conformal sets are averaged over
  a validation set, so I would expect the DTC to hover
  around 0 if the method is well calibrated (but
  positive and negative numbers should be equally
  likely).
- Typo in Fig. 2a: The bottom location should say
  "Set of all possible actions"

**Quality Of The Limitations Section:**

2

**Questions For Rebuttal:**

Can the authors describe the correction procedure and make sure to include in the final version, if accepted?

Can the authors elaborate on how often the algorithm outputs empty sets in practice and why is that a fair comparison with standard conformal prediction?

**Robotics Focus:**

3

**Summary Of Paper:**

This paper addresses the problem of developing a conformal prediction approach for multi-task classification problems, which occur commonly in perception tasks, e.g., in the case of autonomous driving. The authors combine a symbolic knowledge graph with existing conformal prediction methods, coupled with a correction procedure to account for accumulating errors over the multiple tasks. Evaluation is provided over the ROAD and Waymo/ROAD++ datasets.

**Summary Of Recommendation:**

The paper is well written overall, and the approach makes sense. I have a few concerns, which are outlined in the review and in the rebuttal questions.

---

### Official Review · Reviewer_8pRV · 2024-07-17

**Originality:** 3
**Technical Quality:** 4
**Clarity Of Presentation:** 4
**Potential Impact:** 3
**Recommendation:** 3
**Confidence:** 3

**Review:**

The following summarizes strength and weeknesses of the paper.

### Strength
1) The proposed method for semantic-aware CP is novel. The provided mathematical proofs strenghten the generality of the approach.
2) The manuscript is well structured, the notation is well defined and the experimental setup is well-suited for the evaluation of the approach.
3) The results are produced using standard CP metrics, where KRPS outperforms the baseline scoring functions throughout all metrics.
4) The commitment to publish the code upon acceptance is commendable.

### Weaknesses
1) The authors mention that KRPS signalizes knowledge graph gaps or corner cases by producing empty prediction sets. I don't see this reflected in the results, neither in the quantitative nor qualitative evaluation. A discussion and qualitative examples would be helpful.
2) The evaluation assumes a given knowledge graph, however building knowledge graphs can be difficult in practice. A study of how the quality of the knowledge graph influences the results would be helpful to understand the dependency.

**Quality Of The Limitations Section:**

3

**Questions For Rebuttal:**

In addition to the weaknesses above, I found the following language issues:
* L.202: The validation set is split into 3 sets, for validation, calibration and testing. Details on how these splits have been designed would facilitate reproduction of results.
* Fig. 2 a): locations -> actions
* L.218: with respect to s -> with respect to
* L.237: location classification, location classification -> location classification, action classification

**Robotics Focus:**

3

**Summary Of Paper:**

This work proposes Knowledge-Refined Prediction Sets (KRPS), an extension of conformal prediction (CP) for multi-task classification. Specific focus is on the semantic consistency across tasks, which has not been studied before, while maintaining CP's marginal coverage. The presented approach leverages a knowledge graph, that is derived from common sense and the dataset labels, to generate a semantically consistent prediction set from multiple network outputs. The approach is evaluated on two datasets, ROAD and Waymo/ROAD++, using the standard CP metrics. Additionally, a new metric is introduced to evaluate semantic consistency, termed conditional semantic consistency metric.

**Summary Of Recommendation:**

The work introduces a novel approach for conformal predictions in multi-task classification, demonstrating convincing results on two datasets. Despite its strength, I have noted several areas of concerns that I would like the authors to address in their rebuttal.

---

### Official Review · Reviewer_gYc2 · 2024-07-20
**Well-executed paper despite limitations**

**Originality:** 3
**Technical Quality:** 4
**Clarity Of Presentation:** 4
**Potential Impact:** 3
**Recommendation:** 3
**Confidence:** 5

**Review:**

Strength:

1. Ensuring consistency among prediction sets in the multi-tasked setting is a very important topic. KRPS is a natural and intuitive application of cascaded CP for AV perception. The method is well executed.

2. The paper is written cleanly and easy to read.

3. The quantitative results are quite strong — although I am curious if there is any other baseline that can be considered

Limitation:

1. If I understand correctly, there is no probabilistic guarantee for three sets being semantically consistent w.r.t. each other. I guess this is why DTC is fairly large in some experiment settings.

2. It would be good to visualize the knowledge graph more in the main text. The audience might be curious about how complex it is.

3. The coverage right now is marginal, that the guarantee holds in expectation over the calibration sets. Ideally we would like to get coverage conditioning on a particular calibration dataset.

4. I think this approach can be applied in other settings beyond AV. I know the authors choose to focus on AV, but I think some examples in other domains such as navigation or manipulation would strengthen the paper a lot.

**Quality Of The Limitations Section:**

3

**Questions For Rebuttal:**

Is there any possible baseline that also tries to enforce semantic consistency among the prediction sets? Right now the baseline (standard) is pretty weak since it does not try to enforce the consistency at all, if I understand correctly.

I recommend putting a reference for the "1-softmax" score. I know what you meant but the usual audience might not know. You can refer to it as LAC [1] and definitely add to a formal definition of it in the main text.

[1] Mauricio Sadinle, Jing Lei, & Larry Wasserman. “Least Ambiguous Set-Valued Classifiers With Bounded Error Levels.” Journal of the American Statistical Association, 114:525, 223-234, 2019

**Robotics Focus:**

3

**Summary Of Paper:**

This paper introduces KRPS, Knowledge-Refined Prediction Sets, performing semantic-aware conformal prediction for multi-tasked perception. KRPS applies knowledge graph and cascaded CP to ensure the multiple prediction sets are semantically consistent. The method is validated on two AV perception datasets

**Summary Of Recommendation:**

I recommend acceptance of the paper. Overall it is a nice application of cascaded CP and knowledge graph for an important robotics problem. Despite the theoretical limitation, it shows promising improvement over baseline empirically.

---

### Official Review · Reviewer_Tzck · 2024-07-25
**Not Well Written. Reject.**

**Originality:** 3
**Technical Quality:** 1
**Clarity Of Presentation:** 1
**Potential Impact:** 1
**Recommendation:** 1
**Confidence:** 3

**Review:**

The paper proposes a new approach for uncertainty quantification for joint classification tasks: agent classification, action classification, and location classification. The approach uses a pre-defined semantic consistency knowledge graph, which is sued to eliminate class items from the predicted uncertainty set.

I have three major criticisms of this work:
- The paper is not written well. A simple idea is explained painstakingly and in a very non-intuitive manner.
- The rules in perception cannot be hard coded as the authors are proposing here. For example, in Fig  2b, it shows that the class "Cyclist" can only be at the crossing or in the cycle lane. This, however, is not true in practice. It is very much possible for the cycle to be in the middle of the road. Therefore, the method seems to introduce some systematic error into the proposed CP model. I believe this is a major limitation of this approach of reducing the uncertainty set.
- Even if you could codify these rules, the authors do not comment on the scalability of this approach. How many such rules would one have to codify for an application setting like autonomous driving? Is it even practicable?

Specific points to improve the paper:
- The introduction is too verbose and repetitive. Semantic consistency is mentioned several times. The Introduction mainly markets the work, i.e., it argues that the proposed method produces smaller uncertainty set that is semantically consistent. A good introduction provides an overview of the method; however, this is not available in the introduction. A good introduction also provides the scope of the work. This is not made clear in the introduction leading a reader to conclude that the approach is generally applicable.
- It is not clear what "value()" is in an in-line equation in the statement of Theorem 2.1.
- The notion of "semantic consistency" remains unclear till Section 3 of the paper. The paper argues that the methods is aimed to enforce "semantic consistency." However, it should be made clear if they mean "semantic consistency" in a colloquial sense or in an exact definitional sense. It is only in Definition 3.1 that it becomes clear that the authors have a very specific, rule-based specification of "semantic consistency."
- In Section 3, the paper states "the goal is to sequentially construct the prediction set C_l(X_test) for location classification and C_a(X_test) for action classification, given a C_c(X_test) that was generated by an agent classification CP algorithm."
	- It is unclear: "why sequentially?" or "what is sequential here." Up to this point, there has been no indication of it.
	- Given agent classification set seems like a new problem scoping here. The paper did not mention this in the beginning, i.e., the paper considers some form of a conditional model of prediction class first, and then the rest --- namely action and task. Scoping the problem in the beginning would be helpful to the reader.
	- Given this line, why does the last line state that the KRPS also predicts C_c^{KRPS}(X_test)? Isn't C_c predicted independent of the proposed KRPS method?
- In Definition 3.1, are "a" and "c" generic notations or specific to agent action and agent class? If they are generic, it is better to use different different labels than "a" and "c."
- In Definition 3.1, what does "/" mean in Eq (2)? It is not common a mathematical notation.
- The line after Eq (2): should it be "M_{a->c}" instead of "M_{c -> a}"? This statement is unclear, as Eq(2) speaks about image of Y_a in the mapping M_{a -> c}. How do you deduce the pre-image in C_c is not clear.
- The notation "M_{c->l}(C_c)" on line 153 is not correct. The mapping M_{c->l} is defined from point to set, and not set to set (ref: line 131).

**Quality Of The Limitations Section:**

2

**Questions For Rebuttal:**

The authors are free to address any of the comments. I hope my suggestions will help the authors improve their paper.

**Robotics Focus:**

2

**Summary Of Paper:**

The paper proposes a new approach for uncertainty quantification for joint classification tasks: agent classification, action classification, and location classification. The approach uses a pre-defined semantic consistency knowledge graph, which is sued to eliminate class items from the predicted uncertainty set.

**Summary Of Recommendation:**

The paper is not well written would be my main complaint. As for the other two criticisms I mentioned, it would help the paper and the work become significant and attract attention.

---

### Author Rebuttal · Authors · 2024-08-09

# For All Reviewers

We appreciate your thorough and insightful feedback on our manuscript. We have carefully considered all comments and made revisions to enhance the quality of our paper. Below, we summarize the main changes made to address your concerns, grouped by topic and reviewers. Furthermore, we provide detailed answers for points raised by the reviewers as **separate official comments for each reviewer**.

## Knowledge Graph
-  **Knowledge Graph Construction (Tzck, 8pRV, UhpA)**: We have provided more clarifications about our knowledge graph construction procedure. These updates are detailed in Appendix B of our revised manuscript, with references in the main text in L132, L266.

- **Impact of Knowledge Graph Size (8pRV, UhpA)** : We have conducted a new study, which appears in our revised paper in Appendix B.2, exploring KRPS performance using 50%, 70%, and 90% of the full knowledge graph. The results show that while smaller portions of the knowledge graph result in reduced semantic consistency, KRPS maintains robust statistical coverage across varying graph sizes.

## Empty Sets
- **Handling Empty Sets (8pRV, UhpA)**: We have included a discussion and qualitative examples in Appendix E of our revised manuscript, which is referred to in L266 in the main text, illustrating how KRPS identifies and responds to gaps in the knowledge graph or corner cases by producing empty prediction sets. We compared KRPS to baselines, showing that in these cases, empty sets highlight anomalies, where baselines include the wrong classes due to the lack of semantic consistency.

- **Frequency of Empty Sets (8pRV, UhpA)**: We have reported and analyzed the results of a new experiment we conducted showing the frequency of empty set outputs across various testing scenarios in the revised manuscript in Appendix E, with a reference in L 266 in the main revised manuscript.

## Content and Writing Style

- **Introduction Revision (Tzck)**: We have strived to reduce the verbosity and repetition across the introduction, providing an overview of the method, explicitly defining the scope of the work, and clarifying the concept of semantic consistency and sequentiality early in the text.

We Thank all reviewers for their feedback. We hope our revisions as well as our detailed answers posted as official comments for each reviewer, addressed the points raised in the reviews. We remain at disposal to provide further clarifications.

---

### Decision · Program_Chairs · 2024-09-04

**Decision:**

Accept

**Comment:**

strengths

- Reviewers recognize the paper is addressing an important research question, mentioning that "Ensuring consistency among prediction sets in the multi-tasked setting is a very important topic."
- clear applications:  KRPS is a natural and intuitive application of cascaded CP for AV perception.

weaknesses:

- Reviewer Tzck has concerns about the writing and presentation of this paper.
- "KRPS signalizes knowledge graph gaps or corner cases by producing empty prediction sets" not reflected in results.
- lack of explanation of how knowledge graphs can be built and its downstream effect on the prediction results.

After rebuttal, reviews for this paper remains divergent. However, the strong reject was mainly concerned about the writing and presentation of this paper, and the authors give a comprehensive revision on it. Since this paper is addressing an important research question, and the concerns on writing is non substantial and addressable, I would recommend accepting this work.